# Closed-loop recycling of tough epoxy supramolecular thermosets constructed with hyperbranched topological structure

Junheng Zhang [1,2] ✉, Can Jiang[1], Guoyan Deng[1], Mi Luo[3], Bangjiao Ye[3], Hongjun Zhang [3] ✉, Menghe Miao[4], Tingcheng Li[1] & Daohong Zhang[1] ✉

The regulation of topological structure of covalent adaptable networks (CANs) remains a challenge for epoxy CANs. Here, we report a strategy to develop strong and tough epoxy supramolecular thermosets with rapid reprocessability and room-temperature closed-loop recyclability. These thermosets were constructed from vanillin-based hyperbranched epoxy resin (VanEHBP) through the introduction of intermolecular hydrogen bonds and dual dynamic covalent bonds, as well as the formation of intramolecular and intermolecular cavities. The supramolecular structures confer remarkable energy dissipation capability of thermosets, leading to high toughness and strength. Due to the dynamic imine exchange and reversible noncovalent crosslinks, the thermosets can be rapidly and effectively reprocessed at 120 °C within 30 s. Importantly, the thermosets can be efficiently depolymerized at room temperature, and the recovered materials retain the structural integrity and mechanical properties of the original samples. This strategy may be employed to design tough, closed-loop recyclable epoxy thermosets for practical applications.

Epoxy thermosets that exhibit excellent mechanical and thermal performance have attracted considerable attention in the fields of coatings, adhesives, and structural components[1,2]. However, the covalent crosslinked structure of epoxy thermosets hinder reprocessing and recycling, resulting in substantial economic and environmental problems[3,4]. To overcome these problems, covalent adaptable networks (CANs) provide a pragmatic solution, allowing the fabrication of cross-linked healable and recyclable epoxy thermosets, which can dissociate or reversibly crosslink under certain conditions[5,6]. To date, various dynamic covalent-bond-forming processes, including urethane exchange[7], disulfide exchange[8,9], imine exchange[10,11], boronic ester exchange[12], and transesterification reactions[13,14] have been employed to construct CANs, which can be used to produce reprocessable and chemically recyclable epoxy thermosets. Despite considerable effort, the reprocessing process of CANs often relies on a catalyst, a high temperature and high pressure, resulting in unwanted side reactions and a loss of physical properties[15,16]. Consequently, the construction of CANs that are mechanically strong and thermochemically stable and can be rapidly reprocessed under mild conditions without a catalyst remains challenging. Excitingly, dynamic covalent crosslinking enables the resulting crosslinked epoxy thermosets to depolymerize into the original monomers and oligomers, which can then regenerate the thermosets[17,18]. However, these technologies usually require high temperatures, prolonged reaction times, substantial separation and purification[19,20]. There is a critical need to explore efficient, low-cost recycling technologies to achieve closed-loop chemical recycling through depolymerization into monomers and oligomers, followed by full repolymerization of these monomers and oligomers.

[1]Hubei R&D Center of Hyperbranched Polymers Synthesis and Applications, South-Central Minzu University, Wuhan 430074, China. [2]Guangdong Provincial Laboratory of Chemistry and Fine Chemical Engineering Jieyang Center, Jieyang 515200, China. [3]State Key Laboratory of Particle Detection and Electronics, University of Science and Technology of China, Hefei 230026, China. [4]Department of Mechanical Engineering, The University of Melbourne, Grattan Street, Parkville, Victoria 3010, Australia. ✉e-mail: mcjhzhang@gmail.com; hjzhang8@ustc.edu.cn; daohong.zhang@scuec.edu.cn

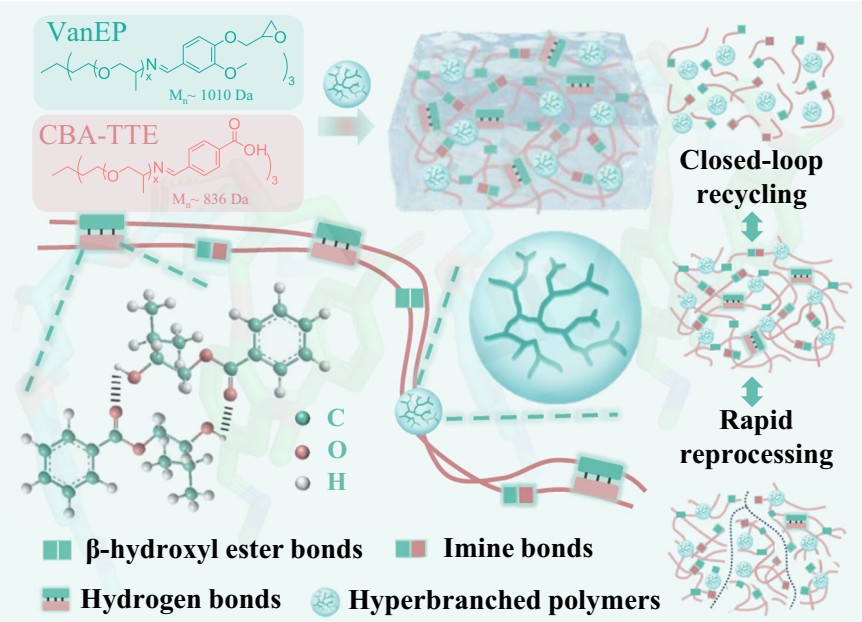

**Fig. 1 | Schematic illustration of the synthesis of epoxy supramolecular thermosets.** Schematic of vanillin-based epoxy resin, vanillin-based hyperbranched epoxy resin and vanillin derivatives bearing phenolic hydroxyl groups; and supramolecular networks prepared from hyperbranched topological structure. The epoxy supramolecular thermosets exhibits high toughness, high strength, closed-loop recycling, and rapid reprocessing simultaneously.

To achieve highly efficient recycling in an economically manner, more recently, several studies have reported simple and energy-saving recycling methods to develop CANs that can be recycled at room-temperature[21,22]. For instance, a supramolecular polyimine with high tensile strength and toughness was developed by introducing dynamic imine bonds and hydrogen bonds, and this material could be depolymerized at room temperature in a THF/HCl mixied solution[22]. Cross-linked polymers formed via reversible amidation chemistry from maleic anhydride and secondary amine monomers can be depolymerized in HCl to realize recover their high-purity in high yield. A supramolecular polysaccharide composed of sodium alginate (SA) and cetyltrimethylammonium bromide (CTAB) was designed, and could be recycled at room temperature via water-induced plasticization[23]. Nevertheless, these room-temperature chemically recyclable polymers exhibited relatively low $T_g$ values. To improve the $T_g$, a recyclable poly-(diketoenamine) with high $T_g$ ($\geq 120\,°C$) was synthesized from triketones and aromatic or aliphatic amines and could be depolymerized at room temperature under strongly acidic conditions (5.0 M $H_2SO_4$)[21]. Furthermore, the existing chemical recycling processes are not yet economical because it is difficult to achieve complete repolymerization of the degradation products. To circumvent such selectivity and quantitative problems, more work is needed to improve the sustainability of the chemistry used to achieve conversion rates of 100% for the degradation product.

In most cases, epoxy thermosets based on CANs have been reported to exhibit unsatisfactory mechanical strength and thermal and oxidative stability due to their conflicting nature that limits their use in structural applications[24]. To overcome this limitation, rigid structures with high-density crosslinking have been widely used for reversible reinforcement. High-density cross-linking can improve the mechanical strength of CANs, but is usually achieved at the expense of reprocessability and recyclability[25]. The incorporation of reversible noncovalent interactions such as multiple hydrogen bonds into CANs can not only endow these polymers with excellent mechanical performance but also provide recyclability and self-healing capacity[26]. In addition to hydrogen bonds, hyperbranched topological structures have also been applied to effectively improve the strength, toughness, solvent resistance and dimensional stability of CANs[27,28].

Hyperbranched polymers have attracted considerable interest due to their unique properties and structural diversity, and have subsequently been used for various applications. Compared with conventional linear polymers, hyperbranched polymers possess intramolecular cavities and abundant functional groups, thus demonstrating excellent potential for strengthening and toughening materials[8,18]. In particular, we fabricated a series of hyperbranched polymers with different hyperbranched topological structures including hyperbranched ionic liquids, hyperbranched epoxy resins and hyperbranched polyesters to tune the polymer properties[17,29–31]. The topologies of these materials resulted in more efficient energy dissipation of the hyperbranched topological structures, which could simultaneously improve the strength and toughness of thermosets, as well as their functions such as flame-retardancy and recyclability[17,29,30]. To make CANs truly suitable as replacements for traditional epoxy thermosets, a different dynamic supramolecular cross-linking approach is needed to design robust CANs that incorporate a hyperbranched topological structure to improve the mechanical durability and thermal stability of dynamic epoxy thermosets.

For these purposes, we hypothesized that if a hyperbranched topological structure that could isomerize to form CANs could be designed, this unique structure could lead to both high strength and excellent toughness. Thus, in this work, we designed a class of epoxy supramolecular thermosets, capable of rapid reprocessing and room-temperature closed-loop chemical recycling. As shown in Fig. 1 and Supplementary Fig. 1, vanillin derivatives bearing phenolic hydroxyl groups (Van-TTE) were prepared by conjugating vanillin with trimethylolpropane tris[poly(propylene glycol), amine terminated] ether. Then, Van-TTE was reacted with epichlorohydrin to obtain vanillin-based epoxy resin (VanEP). Next, vanillin-based hyperbranched epoxy resin (VanEHBP) was synthesized from VanEP and bisphenol A via proton transfer polymerization (Supplementary Fig. 1 and Supplementary Fig. 2). The carboxyl-terminated polyetheramine (CBA-TTE) used as a curing agent was synthesized from 4-formylbenzoic acid and trimethylolpropane tris[poly(propylene glycol), amine terminated] ether. Then, VanEP and VanEHBP were cured with CBA-TTE without a catalyst to form epoxy supramolecular networks, which were synergistically crosslinked by transesterification

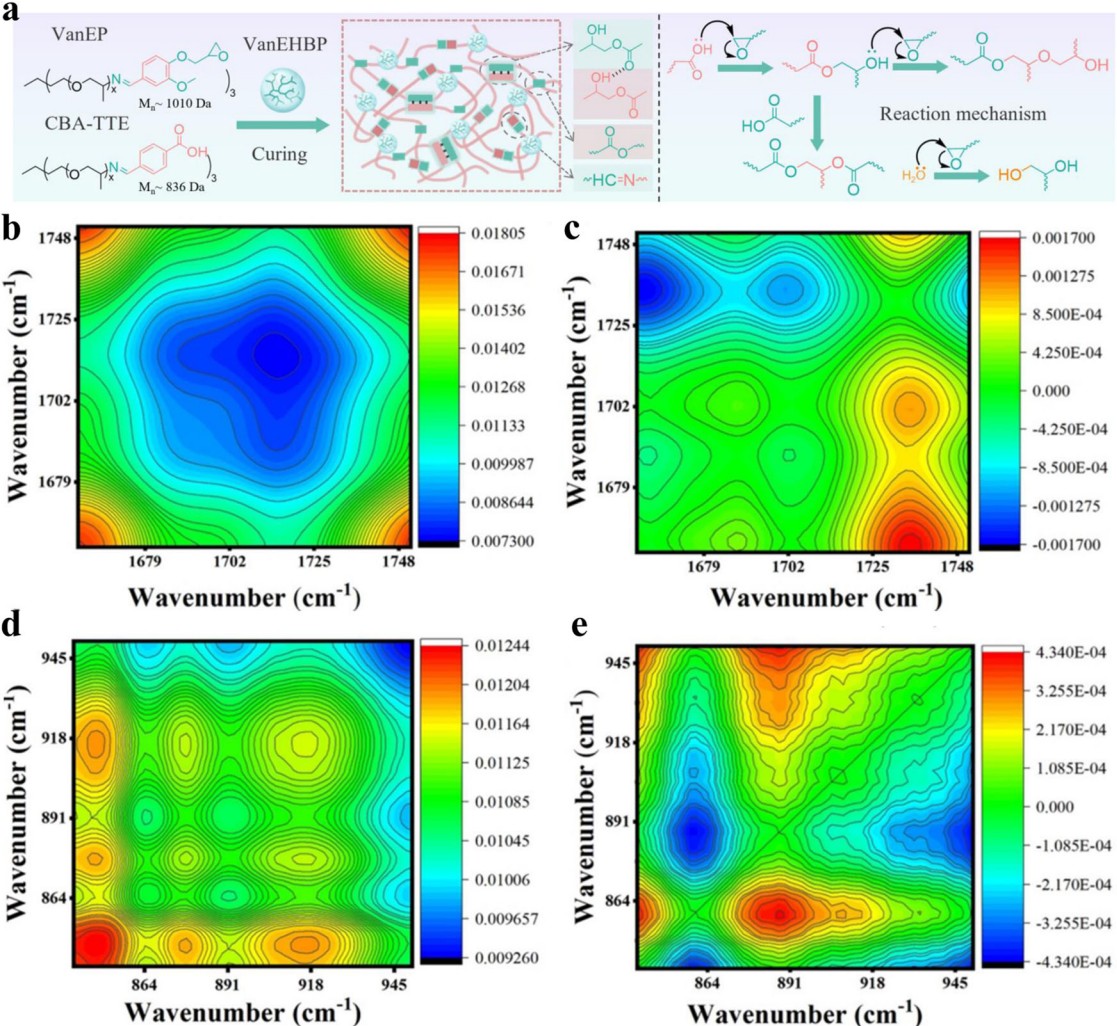

**Fig. 2 | Curing mechanism of epoxy supramolecular thermosets. a** Conceptual illustration of the crosslinking reaction of epoxy supramolecular thermosets (detailed experimental conditions in the Supplementary Methods). Generalized synchronous (**b, d**) and asynchronous (**c, e**) 2D correlation spectra of EN-VanEHBP7 calculated from the temperature-dependent FT-IR spectra from 30 – 120 °C (interval: 3 °C) in the regions of 1660−1750 cm$^{-1}$ vs. 1660−1750 cm$^{-1}$ (**b** and **c**), and 840−950 cm$^{-1}$ vs. 840−950 cm$^{-1}$ (**d** and **e**), red colors represent positive intensities, while blue colors represent negative intensities. Source data are provided in the Source Data file.

and hydrogen bonds. The resultant epoxy supramolecular network (EN-VanEHBP) exhibited excellent mechanical properties, improved creep and chemical resistance, and demonstrated fast reprocessability. More importantly, these epoxy thermosets could be converted into soluble oligomers at room temperature, and then completely regenerated into crosslinked networks without compromising their performance.

## Results

### Construction of a supramolecular network structure via incorporation of a hyperbranched topological structure

The curing process used to prepare the epoxy supramolecular thermosets and the formation of a β-hydroxyl ester are shown in Fig. 2 and Supplementary Fig. 3. As shown, the carboxyl groups of CBA-TTE reacted with the epoxy groups of VanEP and VanEHBP via a ring-opening reaction to form hydroxyl ester bonds. The hydroxyl groups further reacted with epoxy groups via transesterification or formed ester linkages with the carboxyl groups. This curing process of the epoxy supramolecular thermosets was confirmed at different stages by 2D FT-IR analysis. As shown in the synchronous map of these results, the peaks at 1702 and 1725 cm$^{-1}$ were positive, whereas

in the asynchronous map, they were negative. The characteristic peak attributed to epoxy groups at 915 cm$^{-1}$ gradually decreased and disappeared after curing. Additionally, the broad band attributed to carboxyl groups at 1702 cm$^{-1}$ gradually disappeared, followed by the appearance of a new strong peak at 1725 cm$^{-1}$, indicating the formation of ester groups. The curing process used to form the epoxy supramolecular thermosets was further studied by isothermal differential scanning calorimetry (DSC). The curing curves are shown in Supplementary Fig. 4, and the initial curing temperature ($T_i$), peak curing temperature ($T_p$) and total exothermic heat ($\Delta H$) are listed in Supplementary Table 1. All the curves exhibited similar exothermic peaks, which were related to the ring-opening reaction of the epoxy groups. The presence of only one exothermic peak indicated that all the samples were homogeneous reaction systems. The $\Delta H$ value of EN-VanEP (117 J·g$^{-1}$) was much lower than that of EN-DGEBA due to the high viscosity of the EN-VanEP mixtures and thus the need for more heat during curing. With the addition of VanEHBP, $\Delta H$ increased from 117−204 J·g$^{-1}$, and $T_p$ gradually decreased (from 106 °C – 86 °C) with increasing VanEHBP content as a result of the catalytic effect of the abundant VanEHBP hydroxyl groups during the ring-opening reaction.

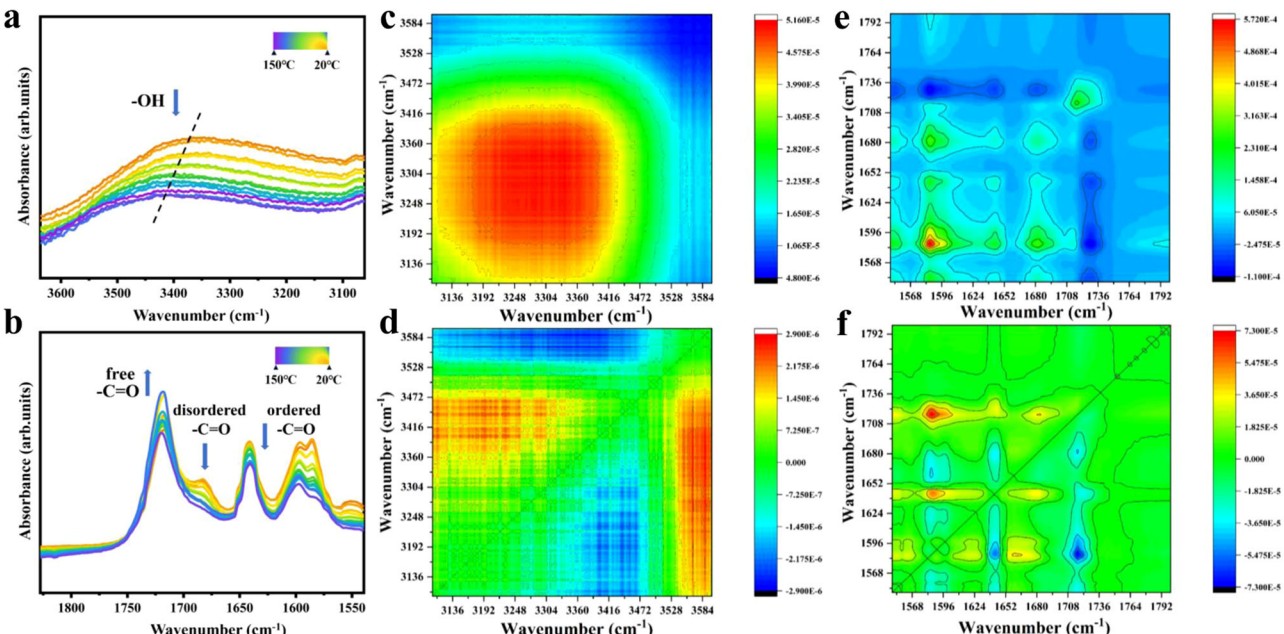

**Fig. 3 | Structural and compositional characterizations of hydrogen-bonding in epoxy supramolecular thermosets. a, b** Variable-temperature infrared spectra of VanEHBP7 upon heating from 20–150 °C (interval: 3 °C): (**a**) 3600-3000 cm$^{-1}$ and (**b**) 1800–1500 cm$^{-1}$. Generalized synchronous (**c, e**) and asynchronous (**d, f**) 2D correlation spectra of EN-VanEHBP7 calculated from the temperature-dependent FT-IR spectra in the regions of 3600-3100 cm$^{-1}$ vs. 3600-3100 cm$^{-1}$ (**c, d**) and 1800–1600 cm$^{-1}$ vs. 1800–1600 cm$^{-1}$ (**e, f**) red colors represent positive intensities, while blue colors represent negative intensities. Source data are provided in the Source Data file.

To study the hydrogen bonding in these materials, FT-IR spectroscopy of these samples was performed, and the results are shown in Fig. 3a–f and Supplementary Fig. 5. The peaks at 1684 and 1641 cm$^{-1}$ were attributed to the stretching vibrations of disordered hydrogen-bonded -C = O and ordered hydrogen bonded -C = O, respectively, while the peak at 1570 cm$^{-1}$ was assigned to the in-plane bending vibration of hydrogen bonded -C = O. In another spectral region, the peak at 3342 cm$^{-1}$ was assigned as the characteristic peak of -OH. In the FT-IR spectra of VanEHBP from 20–150 °C (Fig. 3a, b), the intensity of the peaks attributed to disordered hydrogen-bonded -C = O (1684 cm$^{-1}$) and ordered hydrogen bonded C = O (1641 cm$^{-1}$) gradually decreased, while the intensity of the H-bond intensity of the free -C = O group at 1717 cm$^{-1}$ gradually increased. At the same time, the -OH characteristic peak shifted to a higher wavenumber. Two-dimensional (2D) correlation analysis of the FTIR spectra from 20–150 °C was further performed, and the synchronous and asynchronous maps are shown in Fig. 3c–f. Four correlation cross peaks, namely, Φ (1728, 3364), Φ (1728, 3306), Φ (1652, 3364), and Φ (1652, 3306) were apparent in the synchronous spectrum but absent in the asynchronous spectrum, suggesting the presence of H-bonds in the dynamic networks[32,33].

Regarding the chemical structures of VanEHBP and CBA-TTE, the C = O groups of CBA-TTE and the -OH groups of VanEHBP can for-intermolecular hydrogen bonds (Fig. 4a). To better understand the supramolecular network structure of the thermosets, molecular dynamics (MD) simulations were conducted to elucidate the cross-linking structures of EN-VanEP and EN-VanEHBP (Fig. 4b, c). The average cohesive energies per polymer chain for EN-VanEP and EN-VanEHBP were calculated to be 956.9 kJ·mol$^{-1}$ and 914.2 kJ·mol$^{-1}$ respectively. More specifically, the number of H-bonds in EN-VanEHBP (151 per 2480 CBA-TTE units) was much greater than that in EN-VanEP (74 per 2480 CBA-TTE units), indicating that the intermolecular interactions in EN-VanEHBP are much stronger than those in EN-VanEP. Compared with that of EN-VanEP, the hyperbranched topological structure of VanEHBP favored the formation of H-bonds and thus induced more efficient H-bonding interactions and denser H-bonds. Thus, these results show that the hydrogen-bonding in the supramolecular network endows EN-VanEHBP with excellent mechanical properties, creep and chemical resistance.

## Influence of hyperbranched resin on the mechanical performance of epoxy supramolecular thermosets and its toughening mechanism

Fig. 5a displays the tensile curves of the epoxy supramolecular thermosets, and the relevant data are presented in Supplementary Table 2. Clearly, the mechanical properties of EN-VanEP are comparable to those of EN-DGEBA because the structures of VanEP and DGEBA are similar. The incorporation of VanEHBP simultaneously improved the mechanical properties and $T_g$ of the epoxy supramolecular thermosets. As the content of VanEHBP increased, the mechanical performance of the epoxy supramolecular thermosets first increased, reached a maximum at 7 wt % VanEHBP, and then decreased. The tensile strength, toughness and impact strength of EN-VanEHBP7 were 104.5 MPa, 3.58 MJ·m$^{-3}$ and 57.0 kJ·m$^{-2}$, respectively, which were 43.1%, 109.4% and 128.0% greater than those of the cured EN-VanEP (73.0 MPa, 1.71 MJ·m$^{-3}$ and 25.0 kJ·m$^{-2}$, respectively). Figure S6 and Supplementary Table 3 show the TA curves of the epoxy supramolecular thermosets, and the corresponding data are listed in Supplementary Table 3. As shown, the char yield at 700 °C of EN-VanEHBP7 is much higher than that of EN-DGEBA. It is due to the C = N bonds promote the char formation to form a nitrogen-rich layer at high temperature, resulting in the high char yields of EN-VanEHBP. The effect of VanEHBP on the relative fractional free-volume ($f_r$) of epoxy supramolecular thermosets was investigated by positron annihilation lifetime (PAL) measurements, and the results are presented in Supplementary Table 4. With the addition of VanEHBP which has many intramolecular cavities, an increase in the $f_r$ value of the epoxy network has been achieved. In addition, epoxy chain segments cross-linked with VanEHBP extended in all directions, forming intermolecular cavities, which also contributed to the increase in $f_r$.

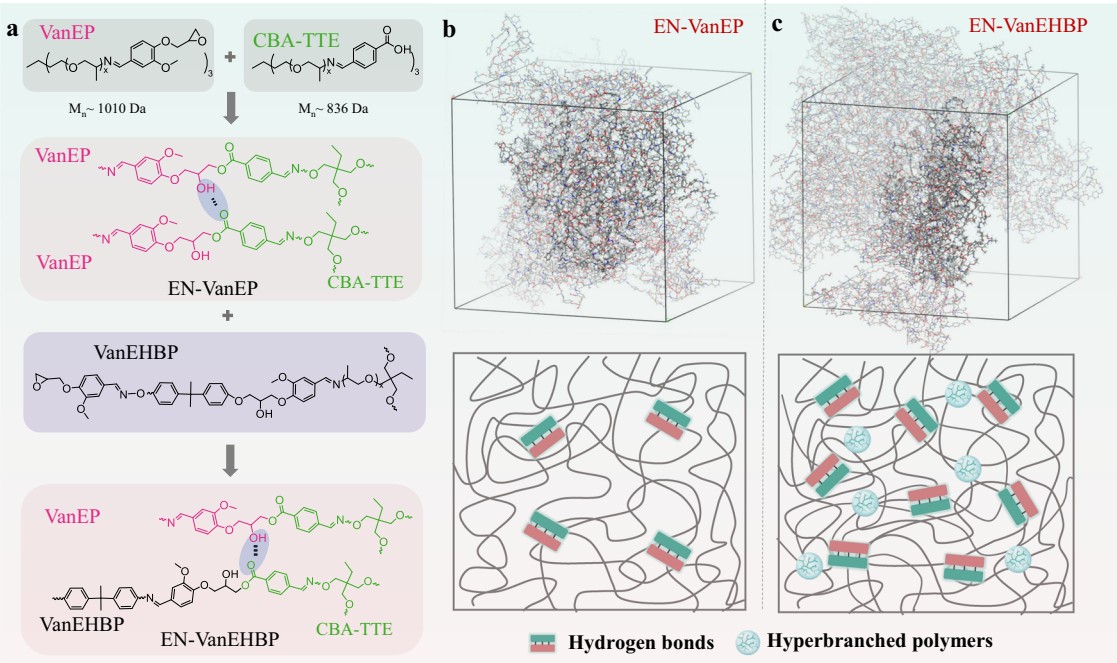

**Fig. 4 | The theoretical investigations of supramolecular network structure.**
**a** Synthetic routes of EN-VanEP and EN-VanEHBP. Snapshots of the all-atom MD simulations of the structures and schematic illustration of the structures of (**b**) EN-VanEP and (**c**) EN-VanEHBP, showing that hydrogen bonds are present at a greater density in EN-VanEHBP than in EN-VanEP (detailed experimental conditions in the Supplementary Methods). Source data are provided in the Source Data file.

However, the intermolecular hydrogen bonds restrict the chain mobility, resulting in a decrease in $f_{rr}$. The PAL results (Supplementary Table 4, Supplementary Fig. 7) indicate that the $f_r$ value decreased first and then increased with increase in VanEHBP content, and EN-VanEHBP7 exhibited the lowest $f_r$ value. According to our previous works[30,34], the mechanical performance of epoxy thermosets toughening with hyperbranched epoxy resin reached their maxima at minimum free volume of the samples. Therefore, the mechanical performance of EN-VanEHBP first increases and then decreases with increasing VanEHBP content, causing the highest tensile stress and toughness to be achieved by EN-VanEHBP7. The dynamic mechanical analysis (DMA) curves of the epoxy supramolecular thermosets are shown in Fig. 5b, c, and the storage modulus ($E_c$), $T_g$ and crosslinking density ($\rho$) are listed in Supplementary Table 2. DGEBA exhibited greater reactivity with the curing agents than did VanEP, thus, the $T_g$ of EN-DGEBA was greater. The introduction of trimethylolpropane tris[-poly(propylene glycol), an amine-terminated ether, as flexible aliphatic segment in VanEP decreased the rigidity of the epoxy supramolecular thermosets, thereby reducing $T_g$. Upon the addition of VanEHBP, the $E_c$ value and the crosslinking density first increased and then decreased. When 7 wt % VanEHBP was added, the $E_c$ reached 2.8 GPa, and when the amount of VanEHBP was increased to 10 wt %, the $E_c$ decreased to 2.1 GPa. The crosslinking density of EN-VanEHBP first increases and then decreases with increasing VanEHBP content, and EN-VanEHBP7 has the highest crosslinking density. The strong intermolecular interactions caused by VanEHBP increase the crosslinking density of EN-VanEHBP. The epoxy group of VanEHBP has a higher degree of functionality than that of linear VanEP, which also contributes to the increased crosslinking density of EN-VanEHBP. But the non-crosslinkable cavities and flexible chain segments will result in a decrease in crosslinking density of EN-VanEHBP. Thus, EN-VanEHBP7 achieves the highest crosslinking density, further substantiating its excellent mechanical performance. Intermolecular hydrogen bonding with VanEHBP and FAT increased the crosslinking density of epoxy supramolecular thermosets. However, the presence of non crosslinkable cavities and flexible chain segments decreased the

crosslinking density[34,35]. Thus, the $T_g$ of EN-VanEHBP first increased and then decreased as the VanEHBP content gradually increased. SEM images of the fracture surfaces of EN-DGEBA, EN-VanEP and EN-VanEHBP7 are presented in Fig. 5d–f. The smooth surfaces in the micrographs indicate brittle fracture of EN-DGEBA and EN-VanEP, while the rough, irregular surfaces of EN-VanEHBP7 suggest significant plastic deformation during ductile fracture. Figure 5g shows the AFM micrographs of the EN-VanEHBP7 fracture surface, including the height image, phase image, and three-dimensional image with dimensions of 5 μm × 5 μm. Only one phase was observed in the AFM topographical image (Fig. 5g). The mapping of the phase image and three-dimensional image showed no phase separation.

The increased mechanical performance of EN-VanEHBP can be explained by the supramolecular networks: (Fig. 5h) (i) The hydrogen bonding interactions effectively stiffened and strengthened the materials. The reversible hydrogen bonds dissipated energy and redistributed stress before failure, thus endowing EN-VanEHBP with high ductility and toughness. (ii) The deformability of the intramolecular and intermolecular cavities provided an efficient pathway to dissipate energy, leading to the stiff yet tough EN-VanEHBP. Moreover, the local free volume associated with hyperbranched crosslinks increased the space available for kink motions, and the crankshaft in the strands enabled secondary relaxation. (iii) The introduction of soft segments reduced the internal stress, and the arms of the hyperbranched crosslinks could easily redistribute the forces, thus reducing the stress concentration. VanEHBP provided greater flexibility between crosslinks and increased the amount of conformational rearrangement during fracturing. Therefore, the as-developed supramolecular structural epoxy thermosets displayed outstanding mechanical performance.

**Dynamics and rapid thermal reprocessing of epoxy supramolecular thermosets from reversible network interactions**
Dual dynamic exchange reactions, including imine exchange and transesterification, of EN-VanEHBP occurred (Fig. 6a). At elevated temperatures, exchange reactions involving imine bonds and

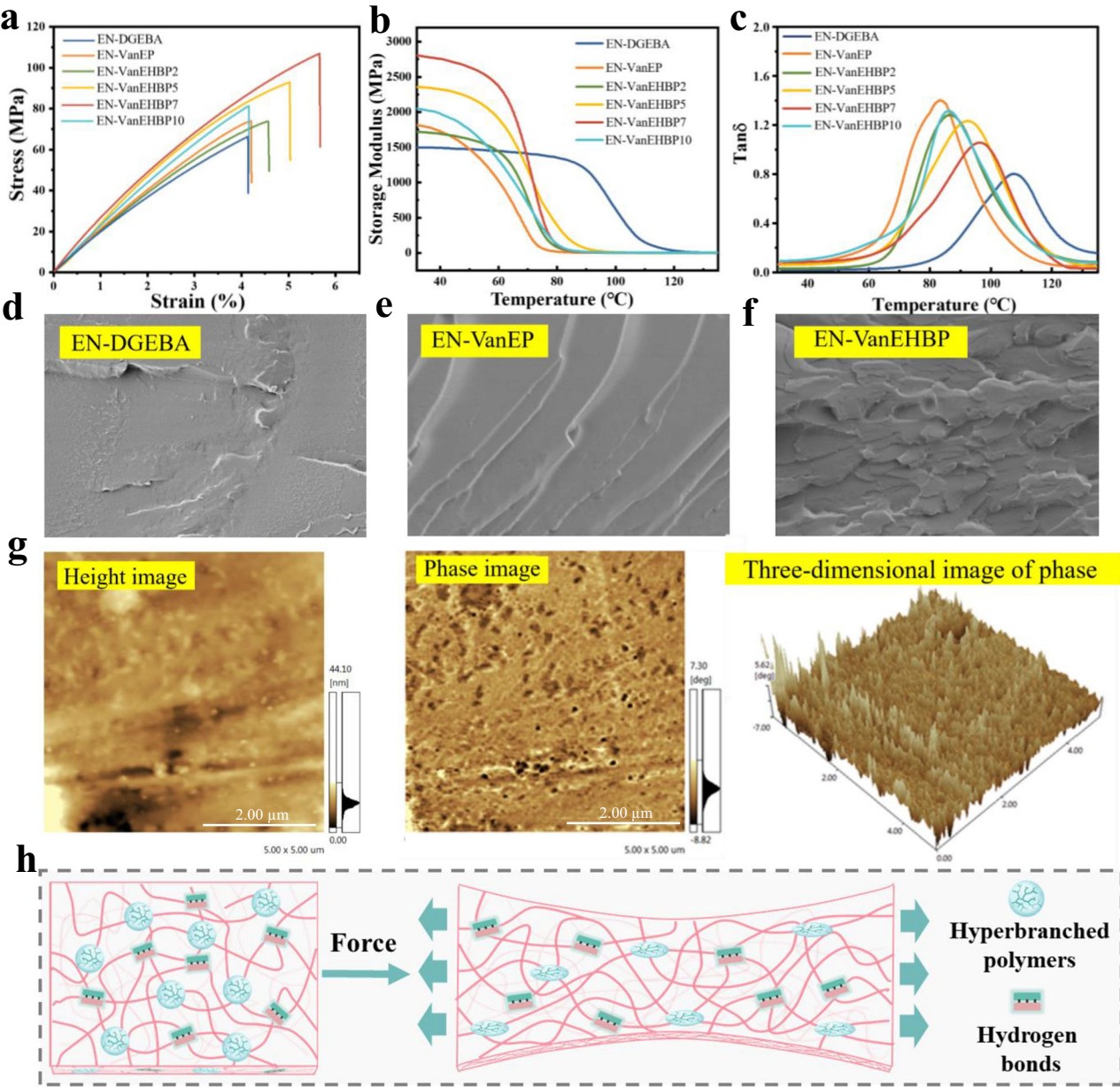

**Fig. 5 | Mechanical properties and toughening mechanism of epoxy supramolecular thermosets. a** Stress-strain curves of epoxy supramolecular thermosets at room temperature (strain rate: 5 mm·min⁻¹). **b** Storage modulus and (**c**) tan curves of epoxy supramolecular thermosets (The curves were recorded in the heating scan from 30–130 °C with a heating rate of 5 °C/min and 1 HZ). SEM images of (**d**) EN-DGEBA, (**e**) EN-VanEP and (**f**) EN-VanEHBP7. Scale bar, 10 μm. **g** AFM images of EN-VanEHBP7 at room temperature. Scale bar, 2 μm. **h** Schematic illustration of the supramolecular networks of EN-VanEHBP upon deformation. The error bars represent the standard deviations of the measured values, *n* = 3 independent samples. Source data are provided in the Source Data file.

transesterification promoted the topological rearrangement of EN-VanEHBP. Although the addition of VanEHBP increased the cross-linking density, which was unfavorable for the exchange reaction, the hydroxyl groups, the high content of ester groups and the short statistical distance between dynamic exchange groups contributed to the mobility and rearrangement abilities of network segments. Thus, owing to the supramolecular networks, EN-VanEHBP was easily reprocessed by compression moulding. Figure 6b–d depict the time-dependent relaxation stress of EN-DGEBA, EN-VanEP and EN-VanEHBP (Fig. 6b) at different temperatures from 100–130 °C. High temperatures facilitated the movement of molecular segments, thus, the relaxation rate increased as the temperature increased. The relaxation time associated with the stress reached 1/e of the initial stress in the stress relaxation curves fitted with the Maxwell model. As shown in

Fig. 6c, absolute stress relaxation was observed for EN-DGEBA, EN-VanEP and EN-VanEHBP7 at high temperatures. For instance, the relaxation times for EN-VanEHBP7, EN-VanEP, and EN-DGEBA decreased from 21.5 s, 39.4 s and 43.3 s to 12.1 s, 14.7 s and 17.3 s, respectively, when the temperature increased from 100 to 130 °C.

The values of the activation energy ($E_a$) of the exchange reaction were calculated using the following equation[36]:

$$In\tau = In\tau_0 + E_a/RT \tag{1}$$

where $\tau_0$ is the relaxation time, and R is the gas constant (8.314 J·mol⁻¹K⁻¹). As shown in Fig. 6c, the curve of In*t* vs. 1000/T was fitted to the Arrhenius law. The $E_a$ values of EN-DGEBA, EN-VanEP, and EN-VanEHBP7 were calculated to be 47.5 ± 2.8 kJ·mol⁻¹,

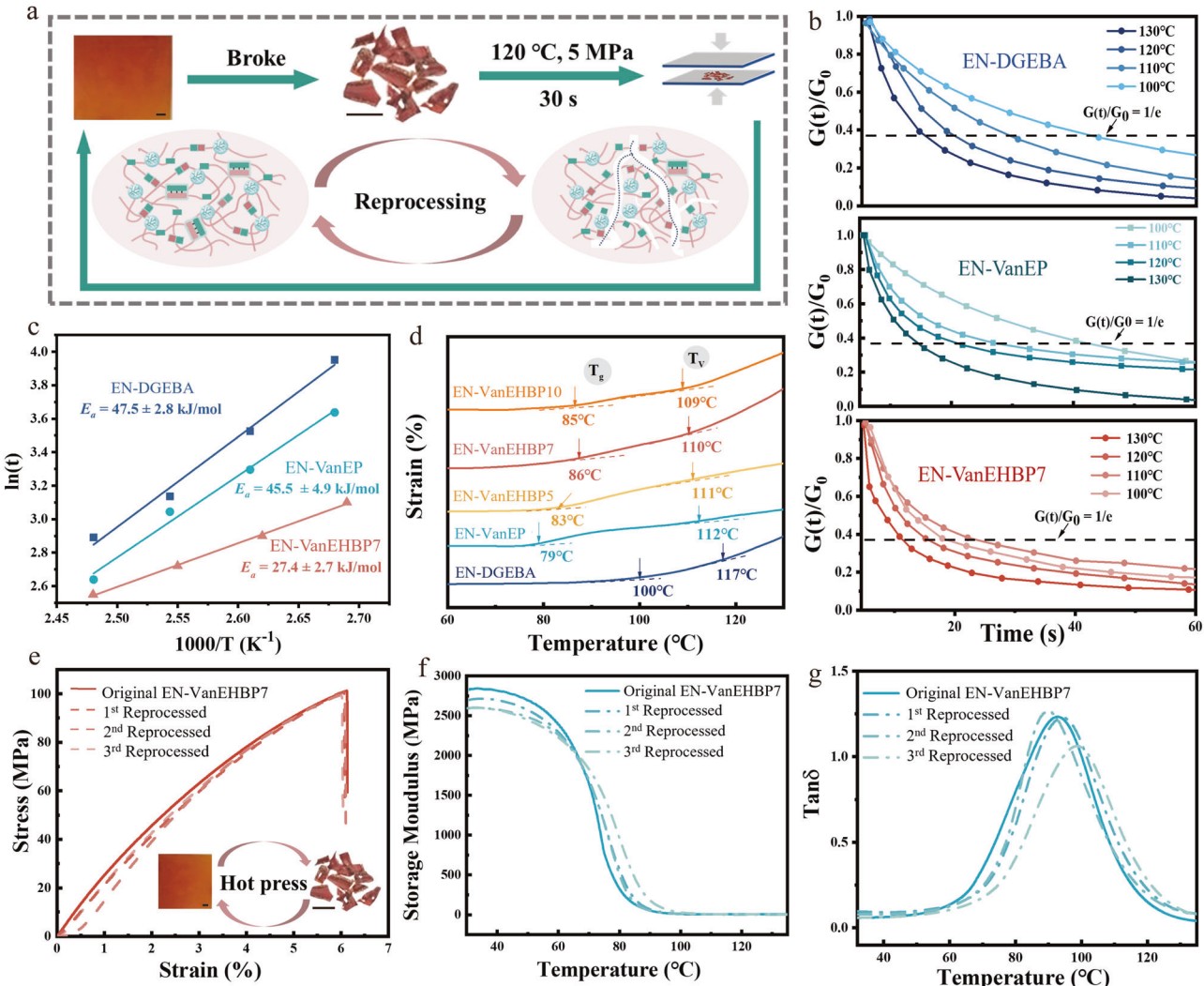

**Fig. 6 | Reprocessing properties of epoxy supramolecular thermosets.**
**a** Rearrangement of EN-VanEHBP7 and the reprocessing process (detailed experimental conditions in the Supplementary Methods). **b** Stress relaxation curves of EN-DGEBA, EN-VanEP and EN-VanEHBP7 at different temperatures, 1/e represents the stress relaxes to 1/e of the initial stress. **c** The fitted curves between In$t$ and 1000/T for epoxy supramolecular thermosets (different colors). The value of $E_a$ is labeled in graph. **d** Temperature dependence of the thermal expansion of the epoxy supramolecular thermosets (different colors). The value of $T_v$ is labeled in graph. **e** Stress-strain curves of the original and reprocessed EN-VanEHBP7 at room temperature (strain rate: 5 mm·min⁻¹). **f, g** Storage modulus and tan δ curves of the original and reprocessed EN-VanEHBP7 (The curves were recorded in the heating scan 30–130 °C with a heating rate of 5 °C/min and 18 HZ). The error bars represent the standard deviations of the measured values, $n = 3$ independent samples. Source data are provided in the Source Data file. Scale bar: 1 cm.

45.5 ± 4.9 kJ·mol⁻¹ and 27.4 ± 2.7 kJ·mol⁻¹, respectively (Supplementary Tables 5). The value of $E_a$ exhibited the same trend as the relaxation rate. As expected, the high content of dynamic bonds in EN-VanEP was conducive to the rearrangement of the molecular segments, and resulted in a lower $E_a$ than that of EN-DGEBA. The $E_a$ of EN-VanEHBP further decreased with the addition of VanEHBP. The topology freezing transition temperatures ($T_v$) of the epoxy supramolecular thermosets were measured via DMA, and the results are shown in Fig. 6d. When the samples were heated above $T_v$, the viscosity significantly decreased, and the samples exhibited Arrhenius-like viscosity variations similar to those of thermoplastic materials. Therefore, when the temperature exceeded $T_v$, the samples exhibited weldability and malleability when the network topology was rearranged[37]. The $T_v$ of EN-VanEP was lower than that of EN-DGEBA, due to the rearrangement of EN-VanEP combined with imine exchange and transesterification. Upon the addition of VanEHBP, the exchange reactions were further promoted and the $T_v$ of EN-VanEHBP gradually decreased with increasing VanEHBP content.

The exchange reaction was accelerated due to the following factors: (1) the catalytic effect of the hydroxyl groups from VanEHBP on transesterification, (2) the higher concentration of ester groups in VanEHBP which was favorable for transesterification, (3) the decrease in the statistical distance between the reactive groups, and (4) the dissociation of the intermolecular hydrogen bonds at high temperatures, which reduced the chain mobility. EN-VanEHBP was found to exhibit a high dynamic exchange rate, excellent reprocessability and a high $T_g$. Moreover, a small model compound study of dynamic imine exchange reactions further demonstrated that the network rearrangement of imine exchange could readily proceed at relatively low temperatures (Supplementary Fig. 7). As a result of the fast exchange reactions, the fragments of EN-VanEHBP7 could be reprocessed into an integral film through hot pressing at 120 °C for 30 s under a pressure of 5 MPa (Fig. 6a). The mechanical and dynamic mechanical properties of the reprocessed samples are shown in Fig. 6e, f, Supplementary Fig. 9 and Supplementary Tables 5–8. The recovery efficiency in terms of tensile strength was almost 100%. Similarly, the storage modulus and $T_g$ were close to those of the original samples after multiple reprocessing cycles,

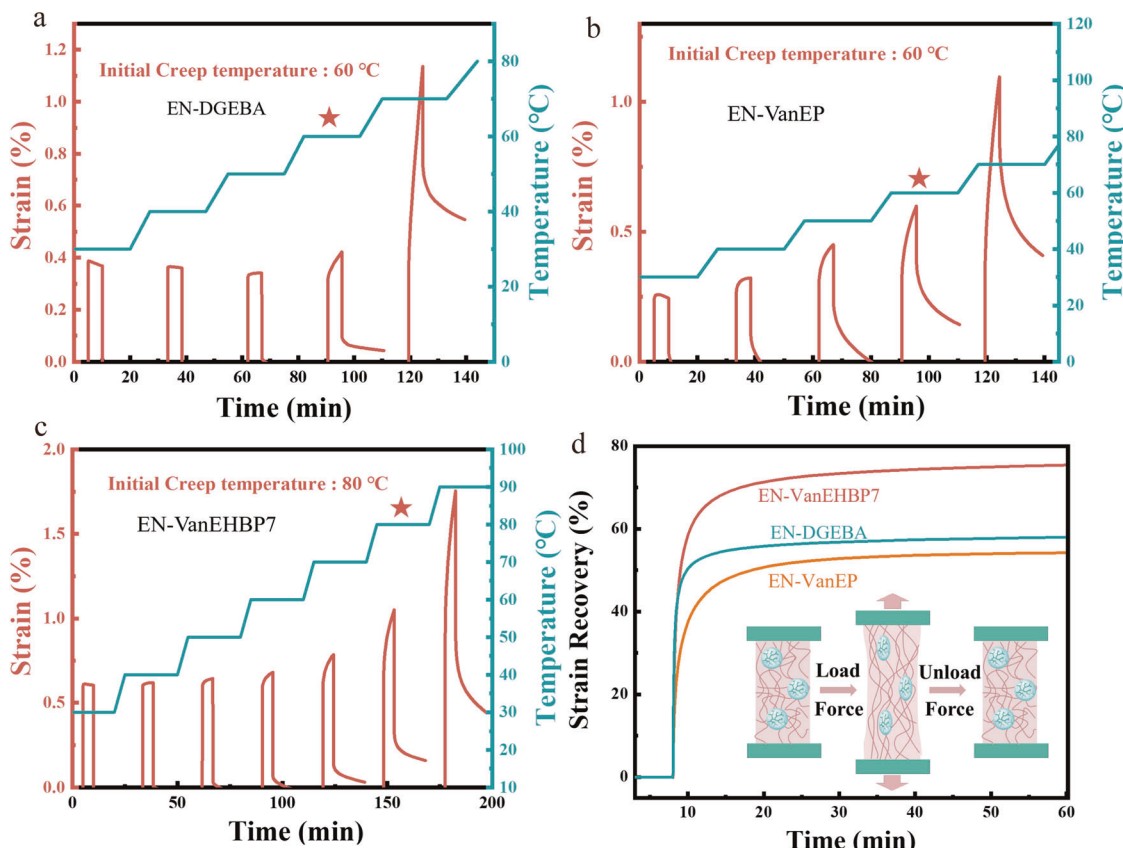

**Fig. 7 | Creep resistance of epoxy supramolecular thermosets.** Creep TTS curves of (**a**) EN-DGEBA, (**b**) EN-VanEP, (**c**) EN-VanEHBP7 and (**d**) creep recovery curves of epoxy supramolecular thermosets at 80 °C. (The curves were recorded in the heating scan from 30 to 100 °C at intervals of 10 °C). All error bars represent s.d. of the mean from three independent experiments ($n$ = 3). Source data are provided as a Source Data file.

indicating that EN-VanEHBP7 could be reprocessed well and could retain its original properties (Fig. 6e, f, Supplementary Fig. 9 and Supplementary Tables 6–9). In addition, the imine exchange and transesterification in supramolecular networks contribute to the excellent malleability and reparability of EN-VanEHBP (Supplementary Fig. 10).

Creep resistance is a key factor for the use of EN-VanEHBP as an engineering plastic and structural material. The creep performance of the prepared epoxy thermosets was tested under constant stress at different temperatures (Fig. 7a–d). After removing the stress, the deformation of EN-VanEP and EN-DGEBA fully recovered at temperatures below 60 °C. Almost no creep was observed for EN-VanEHBP7 until the temperature increased to 80 °C (Fig. 7c). Therefore, EN-VanEHBP showed better dimensional stability than did EN-VanEP and EN-DGEBA. To further investigate the solvent resistance, the cured EN-DGEBA, EN-VanEP and EN-VanEHBP7 samples were immersed in different solvents for 72 h at room temperature to examine their solvent resistance (Supplementary Fig. 11 and Supplementary Table 10). All the samples remained unchanged after being immersed in $H_2O$, ethanol (EtOH), tetrahydrofuran (THF) and dimethylformamide (DMF) at 25 °C for 72 h. With the addition of VanEHBP, the weight loss of EN-VanEHBP7 was much less than that of EN-DGEBA and EN-VanEP because the hydrogen cross-linking induced by VanEHBP allowed the EN-VanEHBP polymer chains to be relatively immobile, enabling them to withstand external forces in various environments and thus leading the improved creep and chemical resistance.

## Closed-loop recycling of epoxy supramolecular thermosets using room temperature mechanism based degradation
Because of their unique reversible chemical bonds, dynamic imine bond-based CANs can be degraded into oligomers containing aldehyde and amine groups under mildly acidic conditions (Fig. 8a and Supplementary Fig. 12). As seen in Supplementary Table 10, the swelling of the prepared epoxy thermosets in DMF was more pronounced than that in other solvents ($H_2O$, EtOH and THF). Therefore, we chose DMF as the solvent to combine with an aqueous HCl solution for degradation. The EN-VanEHBP7 samples were immersed in DMF combined with a 0.1 M HCl aqueous solution (HCl aqueous/DMF = 1/10, v/v) at room temperature. After 180 min, the samples were depolymerized, resulting in a transparent orange degradation solution, and then regenerated oligomers were obtained after removal of the solvent. To further study the degradation of EN-VanEHBP7, the degradation solution and original and chemically recycled EN-VanEHBP7 were analysed by FT-IR and NMR (Supplementary Fig. 13 and Supplementary Fig. 14). In the FT-IR spectra (Supplementary Fig. 13), the CH = N peak at 1643 cm$^{-1}$ in the spectrum of the original samples disappeared, and a CH = O peak at 1683 cm$^{-1}$ appeared in the spectrum of the degradation solution. As shown in Fig. 8b, the $^{13}$C NMR spectrum of depolymerized EN-VanEHBP7 had an obvious aldehyde peak at 192 ppm, and the $^1$H NMR spectrum exhibited an aldehyde peak at 9.83 ppm (Supplementary Fig. 14). Moreover, the solid-state NMR spectra of the original and chemically recycled EN-VanEHBP7 exhibited similar patterns, with an imine bond peaks at 150 ppm in the solid-state $^{13}$C NMR spectra (Supplementary Fig. 15), indicating that the chemical structure of the recycled samples was similar to that of the original samples. To further verify the imine depolymerization, 600 mg of EN-VanEHBP7 was immersed in a solution containing 9 mL of DMSO-$d_6$ and 0.9 mL of 0.1 M HCl/DMSO-$d_6$ solution (HCl/DMSO-$d_6$ = 1/50, v/v) at room temperature and the degradation process was monitored by real-time $^1$H NMR (Fig. 8c). The intensity of the CH = N proton peak ($\delta$ = 8.43 ppm) was found to gradually decrease, and a new peak

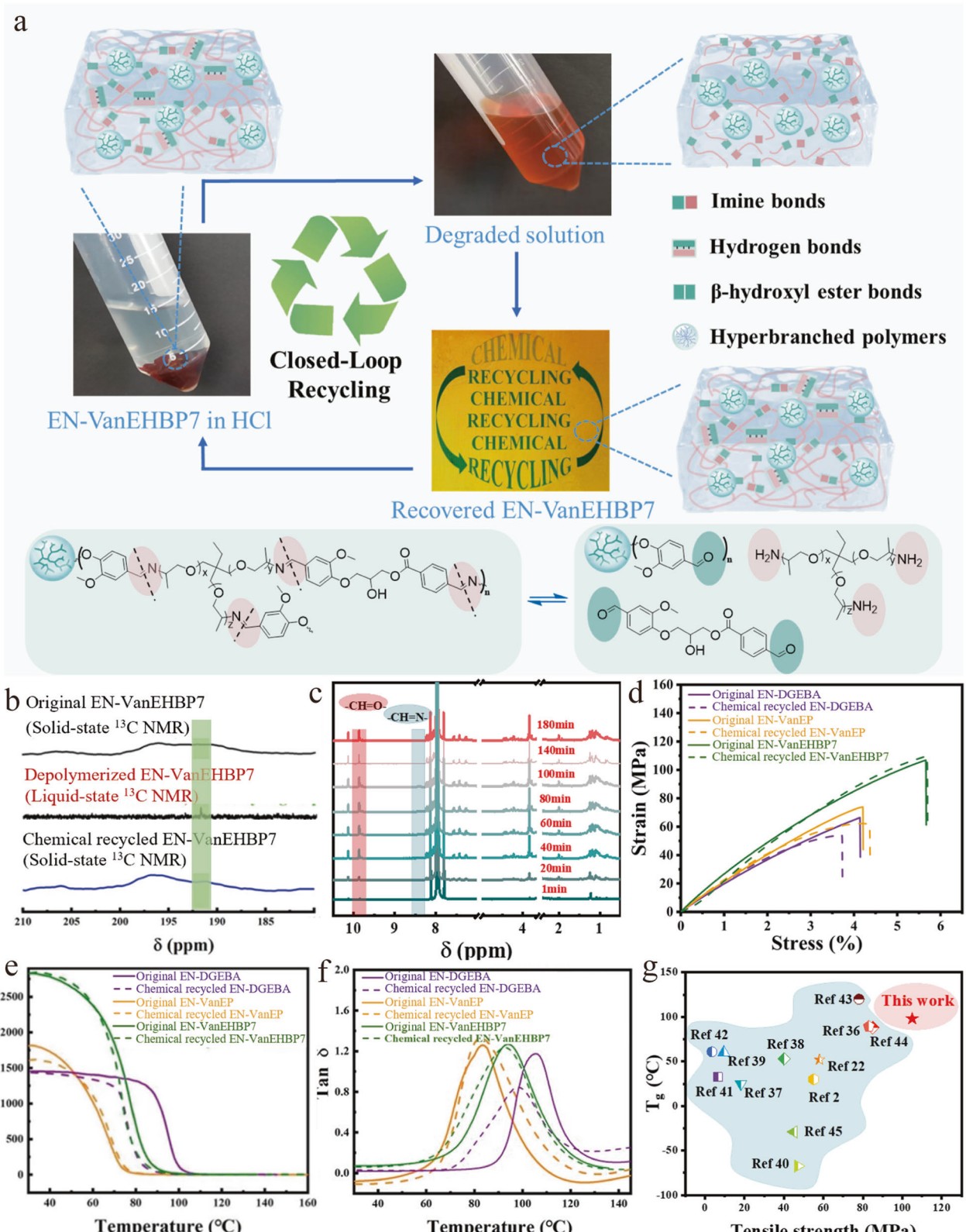

**Fig. 8 | Chemical recycling and repolymerization of epoxy supramolecular thermosets. a** Closed-loop mechanism of EN-VanEHBP7. **b** $^{13}$C NMR spectra of the original, chemically recycled and liquid-state depolymerized EN-VanEHBP7. All spectra were performed on an 100 MHz solid-state NMR spectrometer. **c** Real-time $^1$H NMR spectrum of the EN-VanEHBP7 degradation solution with 0.1 M HCl and DMF at room temperature in the mixture of DMSO-$d_6$. **d** Tensile curves at room temperature (strain rate: 5 mm·min$^{-1}$). **e, f** Storage modulus and tan δ of epoxy supramolecular thermosets before and after chemical recycling (The DMA curves were recorded in the heating scan from 30–130 °C with a heating rate of 5 °C/min and 1 HZ). **g** Comparisons of the tensile strength and $T_g$ of polymers that can be chemically recycled at room temperature in this work and in previous studies. The error bars represent the standard deviations of the measured values, $n = 3$ independent samples. Source data are provided in the Source Data file.

attributed to CH = O ($\delta$ = 9.83 ppm) appeared and increased in intensity, indicating the successful dissociation of EN-VanEHBP7 at room temperature. The integrated area of the CH = O peak in the real-time [1]H NMR spectrum was calculated, and the results indicated that the aldehyde content increased and reached equilibrium after 100 min (Supplementary Fig. 16). The original sample and the chemically recycled sample exhibited similar structure, demonstrating successful room temperature closed-loop recycling of EN-VanEHBP7. The tensile curves, storage modulus and tan $\delta$ vs temperature curves of original and regenerated epoxy thermosets are shown in Fig. 8d–f. The tensile curves of chemically recycled EN-VanEHBP7 were very similar to that of the original sample, with a strength recovery efficiency of nearly 100%. The storage modulus and $T_g$ of the chemically recycled EN-VanEHBP7 were also almost the same as those of the original samples, exhibiting excellent room-temperature closed-loop recyclability. Figure 8g shows that among all the samples, EN-VanEHBP7 exhibited the highest tensile strength and could be recycled at room temperature[2,22,36–45].

## Discussion

In summary, we developed a strategy for preparing strong and tough epoxy thermosets with supramolecular networks by introducing a vanillin-based hyperbranched epoxy resin (VanEHBP). This unique supramolecular network enabled rapid and efficient reprocessing and room-temperature closed-loop chemical recycling of epoxy supramolecular thermosets. The obtained thermosets exhibited much greater tensile strength, toughness and impact strength than traditional petroleum-based products. The supramolecular networks underwent fast stress relaxation and could be reprocessed at 120 °C within only 30 s, with nearly 100% recovery of the mechanical performance after multiple reprocessing cycles. The supramolecular networks exhibited enhanced resistance to deformation and thus improved creep and chemical resistance during service. The epoxy supramolecular thermosets could be chemically fragmented into immediately reusable monomers at room temperature, and the obtained fragment mixture could be reused for recrosslinking to reconstruct epoxy thermosets with nearly 100% material efficiency without losing their original mechanical properties. This work discribes a robust supramolecular crosslinking strategy that affords hyperbranched topological structures for designing energy-efficient and fully closed-loop recycled polymeric products.

## Methods
### Materials
Vanillin (Van, 99.0%), 4-formylbenzoic acid (p-CBA, 98%), trimethylolpropane tris[poly(propylene glycol), amine terminated] ether (TTE, $M_n$ = 440 g·mol$^{-1}$), tetrabutylammonium bromide (99%), sodium hydroxide, bisphenol A (BPA, 99%), were purchased from Shanghai Macklin Biochemical Co., Ltd. Methanol (≥99.7%), ethanol (≥99.7%), ethylacetate (≥99.5%), epichlorohydrin (ECH, ≥99.5%), tetrahydrofuran (THF, ≥99.5%), N,N-dimethylformamide (DMF, ≥99.5%) and hydrochloric acid (HCl, 36.0−38.0%) were purchased from Sinopharm Chemical Reagent Co., Ltd. Diglycidyl ether of bisphenol-A (DGEBA) epoxy resin (a bisphenol epoxy with epoxide equivalent weights 196 g/eq, ≥99.5%) was obtained from Baling Petrochemical Company, Inc. China. All chemicals were used as received unless otherwise discribed.

### Synthesis and characterization of vanillin-based hyperbranched epoxy resin
The synthetic route to prepare vanillin-based hyperbranched epoxy resin (VanEHBP) is shown in Fig. 1 and Supplementary Fig. 1. First, vanillin (22.82 g, 0.15 mol), TTE (22.00 g, 0.05 mol) and 100 mL of ethanol were poured into a 250 mL three-necked flask and mixed evenly under magnetic stirring at room temperature for 30 min. Ethanol was removed using a rotary evaporator and an orange solid Van-TTE was obtained. Then, Van-TTE (42.12 g, 0.05 mol) and

epichlorohydrin (238.80 g, 2.50 mol) were mixed and heated at 110 °C with stirring for 3 h. The above mixture was dissolved in 600 mL of ethyl acetate and reacted for 4 h in an ice bath with the dropwise addition of 120 g of 40 wt % NaOH solution. Finally, the mixture was filtered, washed with water, and dried with anhydrous sodium sulfate, and a yellow liquid (VanEP) was obtained with a yield of 76.0% after rotary evaporation. The epoxy value of VanEP was determined by titration to 0.27 mol·100 g$^{-1}$ according to ASTM D1652. FT-IR (KBr, cm$^{-1}$): 1645 cm$^{-1}$ (-CH = N-), 915 cm$^{-1}$ (C-O-C). [1]H NMR (400 MHz, DMSO-$d_6$) ($\delta$, ppm): 8.19 (s, 1H, -CH = N-), 7.54-7.42 (dd, 2H, Ar-H), 7.34-7.17 (dd, 2H, Ar-H), 7.00-6.90 (dd, 2H, Ar-H), 5.07 (s, 2H, -CH$_2$-O-), 4.45 (dd, 1H, -O-CH$_2$-), 4.32 (dd, 1H, -O-CH$_2$-), 4.05-4.00 (dd, 1H, -O-CH$_2$-), 3.96-3.91 (dd, 1H, -O-CH$_2$-), 3.86 (s, 3H, -OCH$_3$), 3.79-3.76 (m, 1H, -O-CH$_2$-), 3.44-3.37 (dd,1H, -CH-), 3.23 (m, 2H, -CH- in oxirane), 2.89-2.83 (m, 2H, -CH$_2$- in oxirane), 2.74-2.67 (m, 2H, -CH$_2$- in oxirane), 2.00 (s, 2H, -CH$_2$-), 1.24-1.12 (m, 3H, -CH$_3$), 1.02-0.95 (m, 3H, -CH$_3$). [13]C NMR (400 MHz, DMSO-$d_6$) ($\delta$, ppm): 159.40 (-CH = N-), 153.72, 150.00, 130.68, 125.83, 112.35, 109.75, 75.18, 70.29, 65.91, 59.76, 55.39 (-OCH$_3$), 49.54, 43.52, 29.09, 22.94, 17.09, 7.35. The FT-IR and [1]H NMR spectra of VanEP are shown in Supplementary Fig. 17, and the [13]C NMR spectrum of VanEP is shown in Supplementary Fig. 18.

Then, vanillin-based hyperbranched epoxy resin (VanEHBP) was synthesized from VanEP and bisphenol A via proton transfer polymerization. First, BPA (6.84 g, 0.03 mol) and tetrabutylammonium bromide (0.55 g, 0.0017 mol) were dissolved in 20 mL of THF at 50 °C in a 150 mL four-necked round bottom flask with a stirrer and a reflux condenser under a nitrogen atmosphere. After complete dissolution by stirring, a mixture of VanEP (31.92 g, 0.03 mol) and 30 mL of THF was added dropwise to the flask. After the addition of VanEP, the reaction proceeded at 50 °C for 6 h. During the reaction, the molecular weight and epoxy value were measured every 3 h. After the reaction was completed, the resultant product, a yellow VanEHBP liquid with a yield of 98.2 %, was obtained after THF was removed using a rotary evaporator. FT-IR (KBr, cm$^{-1}$): 3524 cm$^{-1}$ (-OH), 1645 cm$^{-1}$ (-CH = N-), 915 cm$^{-1}$ (C-O-C). [1]H NMR (400 MHz, DMSO-$d_6$) ($\delta$, ppm): 8.19 (s, 1H, -CH = N-), 7.51-7.48 (dd, 2H, Ar-H), 7.42-7.38 (dd, 1H, Ar-H), 7.20-7.14 (m, 2H, Ar-H), 7.02-6.99 (dd, 2H, Ar-H), 6.72-6.69 (m, 2H, Ar-H), 4.46-4.43 (dd, 1H, -OH), 4.28-4.25 (dd, 1H, -O-CH$_2$-), 4.23-4.21 (dd, 1H, -O-CH$_2$-), 4.11-4.09 (dd, 1H, -O-CH$_2$-), 3.98-3.96 (dd, 1H, -O-CH$_2$-), 3.85 (s, 3H, -OCH$_3$), 3.60-3.57 (m,1H, -O-CH$_2$-), 3.44-3.41 (dd, 1H, -CH-), 3.30 (m, 2H, -CH- in oxirane), 2.85-2.82 (m, 2H, -CH$_2$- in oxirane), 1.74 (m, 3H, -CH$_3$), 1.52 (s, 2H, -CH$_2$-), 1.43−1.36 (m, 3H, -CH$_3$), 0.94 (m, 3H, -CH$_3$). [13]C NMR (400 MHz, DMSO-$d_6$) ($\delta$, ppm): 159.87 (-CH = N-), 156.80, 155.33, 150.00, 130.50, 127.60, 122.88, 114.78, 112.35, 109.58, 75.36, 70.01-70.39, 69.40, 69.28, 69.14, 68.41, 68.25, 67.08, 65.91, 65.91, 58.46 (-OCH$_3$), 55.21, 46.12, 43.52, 41.09, 28.92, 25.19, 13.19, 7.17. The epoxy value of VanEHBP was titrated according to ASTM D1652 (Supplementary Table 11). The FT-IR and [1]H NMR spectra of VanEHBP are shown in Supplementary Fig. 16, the [13]C NMR spectrum of VanEHBP is shown in Supplementary Fig. 19 and the gel permeation chromatography (GPC) traces of VanEHBP is shown in Supplementary Fig. 20 and Supplementary Table 11.

### Synthesis and characterization of a carboxyl-terminated epoxy curing agent containing dynamic imine bonds
4-Formylbenzoic acid (45.04 g, 0.30 mol), TTE (48.00 g, 0.10 mol) and 100 mL of methanol were added to a 250 ml three-necked flask and magnetically stirred for 30 min. Afterward, the methanol was removed using a rotary evaporator and a yellow liquid 4-formylbenzoic acid-TTE (CBA-TTE), was obtained in 98.9% yield. The acid value of CBA-TTE was determined by titration to be 181 mg KOH·g$^{-1}$ according to ASTMD974. FT-IR (KBr, cm$^{-1}$): 1633 cm$^{-1}$ (-CH = N-), 3452 cm$^{-1}$ (-OH), 1696 cm$^{-1}$ (C = O). [1]H NMR (400 MHz, DMSO-$d_6$) ($\delta$, ppm): 10.12 (s, 1H, -COOH), 8.37 (s, 1H, -CH = N-), 8.00-7.97 (m, 2H, Ar-H), 7.81-7.80 (m, 2H, Ar-H), 5.44 (S, 2H, -CH$_2$-O-), 3.52−3.45 (m, 1H, -O-CH$_2$-), 3.30-3.20 (dd, 1H, -CH-

), 1.18 (S, 2H, -CH$_2$-), 1.01-0.92 (m, 3H, -CH$_3$), 0.77-0.62 (m, 3H, -CH$_3$). $^{13}$C NMR (400 MHz, DMSO-$d_6$) ($\delta$, ppm): 169.94 (-COOH), 159.92 (-CH = N-), 142.64, 132.42, 129.68, 126.77, 75.01, 65.62, 53.12, 49.02, 19.15, 17.78, 7.89. The FT-IR and $^1$H NMR spectra of CBA-TTE are shown in Supplementary Fig. 16, and the $^{13}$C NMR spectrum of CBA-TTE is shown in Supplementary Fig. 21.

## Preparation of the epoxy supramolecular thermosets

Epoxy supramolecular thermosets containing different contents of VanEHBP were prepared according to the following process: stoichiometric amounts of VanEP, VanEHBP and CBA-TTE (the molar ratio of the epoxy group to -COOH was 1:1) were mixed with vigorous stirring at room temperature for 5 min. The obtained viscous uniform mixture was poured into molds and cured at 100 °C for 3 h and 120 °C for 3 h. Control samples of DGEBA with CBA-TTE were prepared via the same method for comparison. The different formulations are summarized in Supplementary Table 11.

## Reprocessing of the epoxy supramolecular thermosets

The reprocessed recycling tests were carried out with a plate vulcanizer. The samples of epoxy supramolecular thermosets samples were cut into small pieces and pressed at 120 °C under a pressure of 5 MPa. After cooling to room temperature, reprocessed epoxy supramolecular thermosets were obtained.

## Chemical recycling of the epoxy supramolecular thermosets

The dynamic imine bonds, commonly known as Schiff base linkages, in covalently crosslinked epoxy supramolecular thermosets can be easily hydrolyzed under mildly acidic conditions. The reversible equilibrium of these dynamic imine bonds in an acidic solution enabled the reversibly crosslinked epoxy supramolecular thermosets to be reconstructed on an interface and thus made the epoxy network closed-loop recyclable by decrosslinking and recrosslinking. Typically, 10 g of epoxy supramolecular thermosets sample were immersed in 150 mL of DMF solvent. Next, 15 mL of 0.1 M HCl aqueous solution was added, and the mixture was stirred gently at room temperature. When the samples were completely dissolved, the degradation solution was orange-red and transparent. The degradation solution was transferred to a single port round bottom flask. The DMF and HCl were removed by a rotary evaporator at 100 °C, and then the sample was heated at 100 °C for 6 h. Afterward, the recycled epoxy supramolecular thermosets were obtained.

## Characterization

$^1$H NMR and $^{13}$C NMR spectra were obtained on an Avance III-400 NMR spectrometer (Bruker Corporation, Germany) using DMSO-$d_6$ as the solvent and tetramethylsilane (TMS) as the internal standard. The $^1$H NMR and $^{13}$C NMR spectra were recorded at 400 MHz. Fourier transform infrared (FT-IR) spectra were collected on a Vertex 70 FT-IR spectrometer (Bruker Corporation, Germany) over the range of 4000–500 cm$^{-1}$ using potassium bromide crystals. The epoxy values of the epoxy resins were determined by the hydrochloric acid-acetone method (ASTM D1652). The molecular weight was measured on a Waters 1525 GPC instrument, with DMF as the eluent at 40 °C after calibration with standard polystyrenes. The injection volume was 15 μL, and the flow rate was 0.3 mL·min$^{-1}$. Differential scanning calorimetry (DSC) tests were performed on a DSC 214 (NETZSCH, Germany) instrument under a nitrogen atmosphere at a heating rate of 10 °C·min$^{-1}$ from 30 – 200 °C. Thermogravimetric analysis (TGA) was carried out using a TGA 209 F3 thermogravimetric instrument (NETZSCH Instruments, Germany) under a nitrogen atmosphere at a heating rate of 10 °C min$^{-1}$ from 30 – 700 °C. Creep, stress-relaxation, $T_v$ and dynamic mechanical analyses were carried out on a DMA Q800 (TA Instruments, USA) at a frequency of 1 Hz in tension mode. The tests were conducted using single cantilever mode at temperatures ranging from 30 – 180 °C, and the specimen dimensions were 20.0 × 4.0 × 1.0 mm$^3$ at a heating rate of 3 °C·min$^{-1}$. Stress-relaxation tests were conducted on rectangular specimens (20.0 × 4.0 × 1.0 mm$^3$), which were initially aligned by preloading with a 0.001 N force, followed by thermal equilibration at each test temperature for 5 min. The specimens were stretched by 1 %, and the strain was retained during the test. The variation in the stress decay profile was recorded as a function of time. The creep TTS test was conducted at temperatures between 30 and 100 °C at intervals of 10 °C with specimen dimensions of 20.0 × 4.0 × 0.5 mm$^3$. During the measurement of each isotherm, a constant force of 3.0 MPa was applied for 5 min, followed by a 15 min recovery period. A creep test was conducted at 80 °C on a specimen with dimensions of 20.0 × 4.0 × 0.5 mm$^3$ under a constant force of 3.0 MPa for 60 min and the time-dependent strain profile was recorded. The tensile strength was measured using a universal tester (INSTRON 5966, Instron, Norwood, MA) according to ASTM D638. Toughness ($\tau$) of the samples can be calculated from the area under the stress ($\sigma$)-strain ($\varepsilon$) curves, using the following equation:

$$\tau = \int_{\varepsilon = 0}^{\varepsilon = \varepsilon_{max}} \sigma d\varepsilon \tag{2}$$

where $\varepsilon_{max}$ is the elongation at break.

Each reported value was the average of at least five specimens. The impact strength was measured using an INSTRON CEAST 9050 impact tester (Torino, Italy) with a 25 J pendulum according to ASTM D256. A minimum of ten specimens were tested for each composition. AFM imaging was carried out using an SPM-9700HT atomic force microscope (Shimadzu Enterprise Management Co., Ltd. China) operating in tapping mode. The topographic (height) and phase images were recorded simultaneously under ambient conditions. Samples with a size of 5 μm × 5 μm were scanned using a high-speed scanner. The morphology of the fracture surfaces was examined by scanning electron microscopy (SEM) (TALOS F200X, China Thermo Fisher Scientific) at 20 kV. Prior to observation, the samples were sputtered with a 1–3 nm layer of platinum using a rotary-pumped sputter coater (Hitachi E-1045, Japan). Gas chromatography-mass spectrometry (GC-MS) spectra were obtained on a QP2010 GC-MS system (Shimadzu, Japan) using ethyl acetate as the solvent. Positron annihilation lifetime (PAL) measurements were recorded in air at room temperature by a conventional fast-fast coincidence system with a time resolution of ~210 ps. The positron source was prepared by depositing and drying a[22] NaCl aqueous solution (with an activity of ~500 kBq) onto the central zone (diameter of <2 mm) of Kapton polyimide foil (10 mm × 10 mm × 7.5 μm, Nilaco) and then covering the first foil with another Kapton polyimide foil of the same size. The positron source was sandwiched between two identical samples with dimensions of 10 mm × 10 mm × 1 mm. The PAL spectrum was collected over 4096 channels with a channel width of 13.24 ps/ch. The total counts of 4 × 10$^6$ for each PAL spectrum were collected in 4 h at a counting rate of ~300 counts/s under air atmosphere. To prevent the backscattering of $\gamma$-rays by the PAL spectroscopy scintillators, the two PAL spectroscopy detectors were positioned perpendicularly. The distance between the sample-source-sample set and each lifetime detector was ~20 mm. By using the PATFIT program, all PAL spectra were decomposed into three lifetime components of $\tau_1$, $\tau_2$ and $\tau_3$ ($\tau_1 < \tau_2 < \tau_3$), with corresponding intensities of $I_1$, $I_2$ and $I_3$, respectively. By using a semiempirical equation based on a spherical infinite potential well model (the Tao-Eldrup model), the average radius ($R$) of the free-volume holes were estimated from the o-Ps lifetime $\tau_3$:

$$\tau_3^{-1} = 2\left[1 - \frac{R}{R + \Delta R} + \frac{1}{2\pi} \sin\left(2\pi \frac{R}{R + \Delta R}\right)\right] \tag{3}$$

where $\Delta R$ is the thickness of the electron layer on the surface of the free-volume holes, which is an empirical parameter of 0.1656 nm. The average volume ($V_f$) of free-volume holes, which is usually called the free-volume hole size, was estimated from

$$V_f = 4\pi R^3/3 \qquad (4)$$

The relative fractional free-volume ($f_r$ in %) was defined as

$$f_r = V_f I_3 \qquad (5)$$

where $V_f$ is the average volume of free-volume holes, $I_3$ is the o-Ps intensity.

The cross-linking density ($\rho$) was calculated by the following equation:

$$\rho = \frac{E_d}{3RT_c} \qquad (6)$$

where $T_c$ is the absolute temperature defined as $T_g + 30\ °C$, $E_d$ is the storage modulus at a temperature of $T_c$, and R is the universal gas constant (8.314 J·mol⁻¹·K⁻¹).

The gel fraction of the samples was measured by means of the dissolution method. Epoxy thermosets (~800 mg, $m_O$) were put into a Soxhlet extractor using THF and immersed for 24 h, followed by drying in a vacuum oven at 80 °C until a constant weight ($m_I$) was reached. The gel fraction value was calculated using the following equation:

$$\text{Gel fraction} = \frac{m_1}{m_0} \times 100\% \qquad (7)$$

The samples (~600 mg, $m_i$) were immersed in different solvents at room temperature for 48 h for swelling. Then, the sample was removed and the solvent on the surface was completely removed before the weight ($m_f$) was measured. The swelling rate was calculated using the following equation:

$$\text{Swelling rate} = \frac{m_f - m_i}{m_i} \times 100\% \qquad (8)$$

### All-atom molecular dynamics simulations

**Simulations of the structures of the supramolecular thermosets.** Molecular dynamics were performed using the software package GRO-MACS (version 2021.3). The system was constructed of packmol and contained 10 molecules in a cubic box with an edge size of 20 nm, which is large enough for these molecules. The atomic interactions were parameterized by the Generation Amber Force Field (GAFF), and the MMFF94 charge obtained from Open Babel was applied in the calculations. After energy minimization, the systems were prebalanced in the NPT ensemble with the Berendsen method for 5 ns. Then, the production run was carried out in the NVT ensemble at 300 K with a time step of 1 fs. The temperature of the system was controlled by a Nose-Hoover thermostat ($\tau T = 1\ ps$). After 10 ns of simulation, the hydrogen bond numbers of the systems were analysed by the toolkits of GROMACS.

### Calculation of the cohesive energy

The cohesive energy per chain was defined as the average energy per chain required to separate all the polymer chains in a condensed state into infinite distances from each other. In our systems, the cohesive energy per chain was calculated by the following equation:

$$E_{cohesive} = \frac{\left[\sum_{i=1}^{10} E_{pot}^{isolated}(i) - E_{pot}^{10}\right]}{10} \qquad (9)$$

where $E_{pot}^{isolated}$ is the average potential energy of an isolated polymer chain in vacuum and $E_{pot}^{10}$ is the average potential energy of the condensed system consisting of five polymer chains. The potential energies of the isolated polymer chains were calculated by averaging 20 frames in 100 picoseconds after the equilibrium simulation of 1 nanosecond. For the condensed system, the potential energy was calculated via the same method after the annealing simulation.

### Reporting summary

Further information on research design is available in the Nature Portfolio Reporting Summary linked to this article.

## Data availability

Source data are present. All relevant data are reported in the manuscript and in the associated Supplementary Information. The data are available from the corresponding author upon request. Source data are provided with this paper. The Raw data generated in this study have been deposited in the Figshare database and are available from https://doi.org/10.6084/m9.figshare.25739286. Source data are provided with this paper.

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

## Acknowledgements

This work was financially supported by the Fundamental Research Funds for the Central Universities, South-Central Minzu University [grant numbers CZY23017 (J.H.Z.), CZD24001 (D.H.Z.)], the National Natural Science Foundation of China [grant numbers U23A20691 (D.H.Z.), 11975225 (H.J.Z.), 12275270 (H.J.Z.), 12175232 (B.J.Y.)], the Natural Science Founda-tion of Hubei Province [grant number 2024AFB800 (J.H.Z.)], the Fund for Academic Innovation Teams of South-Central Minzu University [grant number XTZ24012 (D.H.Z.)] the Scientific Research Platforms of South-Central Minzu University (grant numbers PTZ24013 (J.H.Z.), PTZ24012 (D.H.Z.)] and the Innovation Group of National Ethnic Affairs Commission of China [grant number MZR20006 (D.H.Z.)].

## Author contributions

J.H.Z. conceived the project and designed the experiments. C.J. and G.Y.D. carried out the experiments and data analysis. H.J.Z., M.L., B.J.Y., M.H.M. and T.C.L. participated in the discussion of the manuscript and provided constructive suggestions. D.H.Z. supervised the entire study. All authors commented on the manuscript.

## Competing interests

The authors declare no competing interests.
