## [Peer Review File · Nature Communications]

REVIEWER COMMENTS

Reviewer #1 (Remarks to the Author):

This manuscript reported a vanillin-based hyperbranched epoxy supramolecular thermosets, aiming for high-performance and recyclability. The design of epoxy supramolecular thermosets incorporating hydrogen bond, hyperbranched polymer, imine bond, and β -hydroxyl ester bonds is reasonable. However, the mechanism for constructing high-performance and recyclable thermosets appears insufficiently explained. The emphasis on hydrogen and imine bonds is notable, but there is a lack of attention to hyperbranched polymers and β -hydroxyl ester bonds, with the latter not even mentioned in the main text. Additionally, inconsistencies and missing key data, such as Tg values, solvent resistance, and stress relaxation experiments cannot support relevant results. In general, there are significant issues need to be addressed, and the results appear inconsistent. Therefore, I cannot recommend this manuscript for publication in Nature Communications in the present form. Some comments are listed below.

1. On page 4, line 84, provide details on how to construct and control the hyperbranched structure of epoxy resin, including characterization methods. Also, address how to avoid the formation of cross-linked structures.
2. The significance of β -hydroxyl ester bonds is not mentioned in the main text, and there is an error in the chemical structure representation of β -hydroxyl ester in Fig. 3a.
3. The model compounds of dynamic imine exchange reactions have been well studied. This experiment is unnecessary.
4. The Tg results for the original and reprocessed samples appear inconsistent. In the original sample, EN-VAN-HBP-7 has the highest Tg among these samples, whereas after recycling, EN-VAN-HBP-10 exhibits the highest Tg. Why?
5. On page 3, line 57, where the previous work aiming to improve the Tg of recyclable thermosets is mentioned, please include the specific value of Tg at that point for better contextualization."
6. The results of solvent resistance require further elaboration and quantification. Additionally, the discussion in the main text appears inconsistent with Supplementary Table 10.
7. Review and clarify the results of stress relaxation experiments, particularly the apparent drop in stress below 0% for EN-VAN-HBP-7 at 130 °C after 30 s.
8. In results, the sample name starting with "En-", what does this mean?
9. On page 4, line 83, clarify whether VanEP can be categorized as epoxy resin.
10. The term 'FAT' is not explained or illustrated in either the main text or supplementary information

11. References are missing on page 3, lines 51-55, where the previous work is mentioned. Attach the relevant references for clarity.
12. The manuscript contains spelling mistakes, such as "recycledat" on page 3, line 48. A thorough review for such errors is necessary.

Reviewer #2 (Remarks to the Author):

This work constructed closed-loop recyclable tough epoxy thermosets by hyperbranched topological structure. Epoxy thermosets have been widely used in our life due to their excellent thermal and mechanical properties and solvent resistance. However, they are difficult to be recycled or reused due to their permanently cross-linked networks. The preparation of epoxy thermosets with high mechanical performance and efficient recyclability is still a challenge. In this manuscript, authors addressed these issues using one epoxy system, and vanillin-based hyperbranched epoxy resin was synthesized to prepare tough epoxy supramolecular thermosets. The obtained cross-linked epoxy thermosets exhibited outstanding mechanical properties and room-temperature degradation recyclability, and the toughening mechanism and degradation mechanism were studied and discussed in detail. This study provides an easy strategy to develop ultrastrong, tough, and recyclable epoxy thermosets. Moreover, renewable resource vanillin was used, which made this work more sustainable. This article is the result of an up-to-date research, very well designed, containing a large volume of detailed experiments, and the results are very well arranged and organized. Thus, this is an interesting work and I recommend the publication of this manuscript. Some improvement suggestions are as follows.

1. The incorporation of VanEP, CBA-TTE and VanEHBP for epoxy thermosets could be considered as the novel part of this study. The chemical structures and abbreviations of the used materials for the synthesis of VanEP, CBA-TTE and VanEHBP should be provided.
2. The abbreviations of carboxyl terminated epoxy curing agent containing dynamic imine bonds should be consistent, such as CBA-TTE in Figure 3 but FAT in Figure S1.
3. In Figure S6, the sample information should be clearly indicated.
4. Figure 2 displayed FT-IR spectra in the regions of 3600-3000 cm^{-1} and 1800-1500 cm^{-1} , the Temperature-dependent FT-IR spectra of VanEHBP7 in the range of 4000-500 cm^{-1} should also be provided.
5. The detail descriptions of the Positron annihilation lifetime (PAL) measurement method should be added in section of Characterization.
6. Please add the detail recovery efficiency of mechanical properties of different reprocessed and chemical recycled samples in Table S6.
7. In Figure S12, it is beneficial to give the mass variations of epoxy thermosets in different solvent.

Reviewer #3 (Remarks to the Author):

The work reports a recyclable and tough epoxy resin with mediocre performance and less remarkable recyclability in comparable works. The noteworthy aspect lies in the author's conceptualization of supramolecular thermosets, elaborating the positive effect of hyperbranched topological structure on forming H-bonds. But this could be foreseeable in this field. There is no outstanding contribution in CANS and closed-loop recycling of plastics. The innovation of the proposed strategy and material structure is insufficient, falling short to meet the high requirements of 'Nature Communications'.

1. The quality of manuscript requires significant improvement. The overall expression of the whole manuscript is not clear. The structure of the article might consider extensively reorganization (native English writing, scientific and clear expression, article logicity).
2. The proposed strategy is acceptable, but the schematic diagram is simplistic that cannot represent the hyperbranched topological structure described in the manuscript and its potential effect on cycling of epoxy.
3. The introduction section should be extensively revised from the discussion of significant scientific issues in this field.
4. In Line 81, Regarding the paragraph 'As shown in Figure 1, Van-TTE was formed.....', Where is the 'Van-TTE' label in Figure 1? The description in text and figure should be carefully checked.
5. From the tensile stress-strain curve, the modulus observed did not improve with the concentration of HBP (Fig.4a). Why the storage modulus below T_g yet significantly improves from DMA results (Fig.4b)? And the impact toughness of all materials should be supplied.
6. Close-looped cycling is doubtful. Does it specifically refer to the complete recovery of monomers. The molecular weight of the degraded solution should be supplied. In Fig.8, the description regarding the recycling of materials is confused, which should be clearly showed in the figure.
7. In the conclusion section, the claim that the EP thermoset achieved a high toughness of 3.58 MJ/m³. Checking toughness results of 3.58 MJ/m³, and the 'm³' should be 'm²'.
8. Titles in Results and Discussion seem too ordinary, just showing the properties of material or the characterizations. It is suggested to improve the overall article structure in line with the language style of 'Nature Communications'.
9. The 'Abstract' and 'Conclusion' parts need to be rewritten.
10. There are so many spelling mistakes, formatting errors and unscientific statements. Some are listed as follows. The quality of manuscript should be improved at least.

a) Spelling mistakes:

Line 28, 'wok'

Line 89,'excelent'

line 272, "depolymerised"

b) Miss spaces:

Line 32 'thermosetsthat','performancehave';

Line 38 'includeurethane';

Line 44 'andthere';

Line 48 'recycledat'?

Line 72, 'cavitiesand', 'instrengthening'

Line 144 'Theincrease'

Line 148 'andtended'

Line 220 'VanEHBPwhich'.....

c) Others:

In line 148, 'thus' should be 'Thus';

line 122, "simulated" should be "stimulation"

Supporting information:

Such as S1.3: '181 mgKOH·g-1';¹H NMR (400MHZ, DMSO-D6)'

Response to Referee 1

General comments:

This manuscript reported a vanillin-based hyperbranched epoxy supramolecular thermosets, aiming for high-performance and recyclability. The design of epoxy supramolecular thermosets incorporating hydrogen bond, hyperbranched polymer, imine bond, and β -hydroxyl ester bonds is reasonable. However, the mechanism for constructing high-performance and recyclable thermosets appears insufficiently explained. The emphasis on hydrogen and imine bonds is notable, but there is a lack of attention to hyperbranched polymers and β -hydroxyl ester bonds, with the latter not even mentioned in the main text. Additionally, inconsistencies and missing key data, such as Tg values, solvent resistance, and stress relaxation experiments cannot support relevant results. In general, there are significant issues need to be addressed, and the results appear inconsistent. Therefore, I cannot recommend this manuscript for publication in Nature Communications in the present form. Some comments are listed below.

Author response:

We sincerely appreciate your careful reading of our work and the insightful input. Accordingly, we have revised the manuscript to address all the points that were raised. The revised sections are written in blue font, both in the response letter and in the revised manuscript, text and figures. We feel that the revised work is much improved as a result. Please refer to the detailed point by point responses below. We would like to thank you again for considering our work.

Referee comment 1. On page 4, line 84, provide details on how to construct and control the hyperbranched structure of epoxy resin, including characterization methods. Also, address how to avoid the formation of cross-linked structures.

Author response: We sincerely thank you for your valuable suggestion. Controlling the hyperbranched structure of epoxy resin is the key to develop epoxy supramolecular thermosets. The detailed synthesis of vanillin-based hyperbranched epoxy resin has been added in the Supplementary Information. In our experiment, it is important to control reaction temperature, concentration of reagents and rate of reagents addition to construct the hyperbranched structure and avoid the formation of cross-linked structures. The reaction process was monitored by FT-IR, GPC and titration method. The revisions are as below in blue font.

1.2 Synthesis and characterization of vanillin-based hyperbranched epoxy resin

The synthetic route to prepare vanillin-based hyperbranched epoxy resin (VanEHBP) is shown in Figure 1 and Figure S1. First, vanillin (22.82 g, 0.15 mol), TTE (22.00 g, 0.05 mol) and 100 mL of ethanol were poured into a 250 mL three-necked flask and mixed evenly under magnetic stirring at room temperature for 30 min. Ethanol was removed using a rotary evaporator and an orange solid Van-TTE was obtained. Then, Van-TTE (42.12 g, 0.05 mol) and epichlorohydrin (238.80 g, 2.50 mol) were mixed and heated at 110 °C with stirring for 3 h. The above mixture was dissolved in 600 mL of ethyl acetate and reacted for 4 h in an ice

bath with the dropwise addition of 120 g of 40 wt% NaOH solution. Finally, the mixture was filtered, washed with water, and dried with anhydrous sodium sulfate, and a yellow liquid (VanEP) was obtained with a yield of 76.0 % after rotary evaporation. The epoxy value of VanEP was determined by titration to be 0.27 mol 100 g⁻¹ according to ASTM D1652. FT-IR (KBr, cm⁻¹): 1645 cm⁻¹ (-CH=N-), 915 cm⁻¹ (C-O-C). ¹H NMR (400 MHz, DMSO-D₆) (δ, ppm): 8.19 (s, 1H, -CH=N-), 7.54-7.42 (dd, 2H, Ar-H), 7.34-7.17 (dd, 2H, Ar-H), 7.00-6.90 (dd, 2H, Ar-H), 5.07 (s, 2H, -CH₂-O-), 4.45 (dd, 1H, -O-CH₂-), 4.32 (dd, 1H, -O-CH₂-), 4.05-4.00 (dd, 1H, -O-CH₂-), 3.96-3.91 (dd, 1H, -O-CH₂-), 3.86 (s, 3H, -OCH₃), 3.79-3.76 (m, 1H, -O-CH₂-), 3.44-3.37 (dd, 1H, -CH-), 3.23 (m, 2H, -CH- in oxirane), 2.89-2.83 (m, 2H, -CH₂- in oxirane), 2.74-2.67 (m, 2H, -CH₂- in oxirane), 2.00 (s, 2H, -CH₂-), 1.24-1.12 (m, 3H, -CH₃), 1.02-0.95 (m, 3H, -CH₃). ¹³C NMR (400 MHz, DMSO-D₆) (δ, ppm): 159.40 (-CH=N-), 153.72, 150.00, 130.68, 125.83, 112.35, 109.75, 75.18, 70.29, 65.91, 59.76, 55.39 (-OCH₃), 49.54, 43.52, 29.09, 22.94, 17.09, 7.35. The FT-IR and ¹H NMR spectra of VanEP are shown in Figure S16, and the ¹³C NMR spectrum of VanEP is shown in Figure S17.

Then, the vanillin-based hyperbranched epoxy resin (VanEHBP) was synthesized from VanEP and bisphenol A via proton transfer polymerization. First, the weighed BPA (6.84 g, 0.03 mol) and tetrabutylammonium bromide (0.55 g, 0.0017 mol) were dissolved in 20 mL of THF at 50 °C in a 150 mL four-necked round bottom flask with a stirrer and a reflux condenser under a nitrogen atmosphere. After complete dissolution by stirring, a mixture of VanEP (31.92 g, 0.03 mol) and 30 ml of THF was added dropwise to the flask. After the addition of VanEP, the reaction proceeded at 50 °C for 6 h. During the reaction, the molecular weight and epoxy value were measured every 3 h. After the reaction was completed, the resultant product, a yellow VanEHBP liquid with a yield of 98.2 %, was obtained after THF was removed using a rotary evaporator. FT-IR (KBr, cm⁻¹): 3524 cm⁻¹ (-OH), 1645 cm⁻¹ (-CH=N-), 915 cm⁻¹ (C-O-C). ¹H NMR (400 MHz, DMSO-d₆) (δ, ppm): 8.19 (s, 1H, -CH=N-), 7.51-7.48 (dd, 2H, Ar-H), 7.42-7.38 (dd, 1H, Ar-H), 7.20-7.14 (m, 2H, Ar-H), 7.02-6.99 (dd, 2H, Ar-H), 6.72-6.69 (m, 2H, Ar-H), 4.46-4.43 (dd, 1H, -OH), 4.28-4.25 (dd, 1H, -O-CH₂-), 4.23-4.21 (dd, 1H, -O-CH₂-), 4.11-4.09 (dd, 1H, -O-CH₂-), 3.98-3.96 (dd, 1H, -O-CH₂-), 3.85 (s, 3H, -OCH₃), 3.60-3.57 (m, 1H, -O-CH₂-), 3.44-3.41 (dd, 1H, -CH-), 3.30 (m, 2H, -CH- in oxirane), 2.85-2.82 (m, 2H, -CH₂- in oxirane), 1.74 (m, 3H, -CH₃), 1.52 (s, 2H, -CH₂-), 1.43-1.36 (m, 3H, -CH₃), 0.94 (m, 3H, -CH₃). ¹³C NMR (400 MHz, DMSO-d₆) (δ, ppm): 159.87 (-CH=N-), 156.80, 155.33, 150.00, 130.50, 127.60, 122.88, 114.78, 112.35, 109.58, 75.36, 70.01-70.39, 69.40, 69.28, 69.14, 68.41, 68.25, 67.08, 65.91, 65.91, 58.46 (-OCH₃), 55.21, 46.12, 43.52, 41.09, 28.92, 25.19, 13.19, 7.17. The epoxy value of VanEHBP was titrated according to ASTM D1652 (Table S11). The FT-IR and ¹H NMR spectra of VanEHBP are shown in Figure S16, the ¹³C NMR spectrum of VanEHBP is shown in Figure S18 and the gel permeation chromatography (GPC) trace of VanEHBP is shown in Figure S19 and Table S11.

Supplementary Figure 19. Evolution of the molecular weight during polymerization for the syntheses of VanEHBP.

Supplementary Table 11. GPC data for molecular weight determination and epoxy value of VanEHBP

Polymerization time (h)	M_n (g mol ⁻¹)	M_w (g mol ⁻¹)	PDI	Epoxy value (mol 100 g ⁻¹)
3	3519	5806	1.65	0.21
6	5271	9172	1.74	0.18
9	5405	11567	2.14	0.12

Referee comment 2. The significance of β -hydroxyl ester bonds is not mentioned in the main text, and there is an error in the chemical structure representation of β -hydroxyl ester in Fig. 3a.

Author response: We sincerely thank you for carefully reading the manuscript and providing highly valuable guidance. You raised a very important point concerning the significance of β -hydroxyl ester bonds in the supramolecular thermosets. We have corrected the chemical structure representation of β -hydroxyl ester accordingly and added more discussions about the β -hydroxyl ester bonds. The relevant revision in the revised manuscript is as follow in blue font.

Fig. 4. (a) Synthetic routes of EN-VanEP and EN-VanEHBP. Snapshots of the all-atom MD simulations of the structures and schematic illustration of the structures of (b) EN-VanEP and (c) EN-VanEHBP, showing that hydrogen bonds are present at a greater density in EN-VanEHBP than in EN-VanEP.

The curing process used to prepare epoxy supramolecular thermosets and the formation of a β -hydroxyl ester are shown in Figure 2 and Figure S3. As shown, the carboxyl groups of CBA-TTE reacted with the epoxy groups of VanEP and VanEHBP via a ring-opening reaction to form hydroxyl ester bonds. The hydroxyl groups further reacted with epoxy groups via transesterification or formed ester linkages with the carboxyl groups. This curing process of epoxy thermosets was confirmed at different stages by 2D FT-IR analysis. As shown in the synchronous map of these results, the peaks at 1702 and 1725 cm^{-1} were positive, whereas in the asynchronous map, they were negative. The characteristic peak attributed to epoxy groups at 915 cm^{-1} gradually decreased and disappeared after curing. Additionally, the broad band attributed to carboxyl groups at 1702 cm^{-1} gradually disappeared, followed by the appearance of a new strong peak at 1725 cm^{-1} , indicating the formation of ester groups.

Fig. 2. (a) Conceptual illustration of the crosslinking reaction of epoxy supramolecular thermosets. Generalized synchronous (b, d) and asynchronous (c, e) 2D correlation spectra of EN-VanEHBP7 calculated from the temperature-dependent FT-IR spectra in the regions of 1660-1750 cm^{-1} vs. 1660-1750 cm^{-1} (b and c), and 840-950 cm^{-1} vs. 840-950 cm^{-1} (d and e).

Referee comment 3. The model compounds of dynamic imine exchange reactions have been well studied. This experiment is unnecessary.

Author response: We sincerely thank you for your highly instructive comments. Following your suggestion, the discussion of the model compounds of dynamic imine exchange reactions have been moved to the Supplementary Information.

Referee comment 4. The T_g results for the original and reprocessed samples appear inconsistent. In the original sample, EN-VAN-HBP-7 has the highest T_g among these samples, whereas after recycling, EN-VAN-HBP-10 exhibits the highest T_g . Why?

Author response: We thank you for raising this important point. In our experiment, EN-DGEBA, EN-VanEP and EN-VanEHBP7 were reprocessed and chemically recycled to investigate the closed-loop recyclability. We have carefully repeated the measurements and updated our results of T_g , the corrected results are shown below (Figs. 5 b and c, Table S2 and Fig. 6d). As shown, the T_g of EN-VanEHBP first increased, reaching a maximum at 7 wt% VanEHBP, and then decreased.

Fig. 5. (a) Stress-strain curves of epoxy supramolecular thermosets. (b) Storage modulus and (c) tan curves of epoxy supramolecular thermosets. SEM images of (d) EN-DGEBA, (e) EN-VanEP and (f) EN-VanEHBP7. (g) AFM images of EN-VanEHBP7. (h) Schematic illustration of the supramolecular networks of EN-VanEHBP upon deformation.

Supplementary Table 2. Mechanical and thermomechanical properties of the epoxy supramolecular thermosets

Samples	Tensile Strength (MPa)	Toughness (MJ·m ⁻²)	Young's Modulus (MPa)	Impact Strength (KJ·m ⁻²)	E _c (GPa)	T _g (°C)	E _d (MPa)	ρ (×10 ⁻³ mol cm ⁻³)
EN-DGEBA	66.3±5.6	1.51±0.14	1832±334	24.0±2.7	1.5	105	2.9	0.29
EN-VanEP	73.0±4.8	1.71±0.09	1945±245	25.0±3.4	1.8	83	1.3	0.14
EN-VanEHBP2	74.1±3.2	1.80±0.12	1998±281	29.5±3.6	1.7	86	1.3	0.13
EN-VanEHBP5	92.9±3.1	2.66±0.10	2332±256	48.3±3.5	2.4	92	3.0	0.30
EN-VanEHBP7	104.5±3.4	3.58±0.08	2400±214	57.0±3.3	2.8	95	4.6	0.46
EN-VanEHBP10	81.3±3.2	1.81±0.13	2223±273	42.6±3.6	2.1	86	2.8	0.29

E_c is the storage modulus at 30 °C, T_g is the glass transition temperature, E_d is the storage modulus at T_g + 30 °C, and ρ is the crosslinking density.

Fig. 6. (a) Rearrangement of EN-VanEHBP7 and the reprocessing process. (b) Stress relaxation curves of EN-DGEBA, EN-VanEP and EN-VanEHBP7 at different temperatures. (c) The fitted curves between $\ln t$ and $1000/T$ for epoxy supramolecular thermosets. (d) Temperature dependence of the thermal expansion of the epoxy supramolecular thermosets. (e) Tensile strength of the original and reprocessed EN-VanEHBP7. (f) Storage modulus and $\tan \delta$ curves of original and reprocessed EN-VanEHBP7. (g) Comparisons of the tensile strength and reprocessing time of thermosets containing imine bonds in this work and in previous studies.

Referee comment 5. On page 3, line 57, where the previous work aiming to improve the T_g of recyclable thermosets is mentioned, please include the specific value of T_g at that point for better contextualization."

Author response: We fully agree that the specific value of T_g should be mentioned for better contextualization. The corresponding revisions are in blue font.

To improve T_g , a recyclable poly (diketoenamine)s with high T_g (≥ 120 °C) was synthesized from triketones and aromatic or aliphatic amines, which can be depolymerized at room temperature in strong acidic conditions (5.0 M H_2SO_4)²¹.

[21] Christensen, P. R. et al. Closed-loop recycling of plastics enabled by dynamic covalent diketoenamine bonds. *Nat. Chem.* 11, 442-448 (2019).

Referee comment 6. The results of solvent resistance require further elaboration and quantification. Additionally, the discussion in the main text appears inconsistent with Supplementary Table 10.

Author response: We thank you for raising this important point. We have re-run the solvent resistance tests and supplemented discussions related to this point together with Supplementary Figure R10 and Supplementary Table R9 in the revised manuscript. The

relevant revision in the revised manuscript is as follow in blue font.

To further investigate the solvent resistance, the cured EN-DGEBA, EN-VanEP and EN-VanEHBP7 samples were immersed in different solvents for 72 h at room temperature to further examine their solvent resistance (Figure S10 and Table S9). All the samples remained unchanged after being immersed in H₂O, ethanol (EtOH), tetrahydrofuran (THF) and dimethylformamide (DMF) at 25 °C for 72 h. With the addition of VanEHBP, the weight loss of EN-VanEHBP7 was much less than those of EN-DGEBA and EN-VanEP because the hydrogen cross-linking induced by VanEHBP allowed the EN-VanEHBP polymer chains to be relatively immobile, enabling them to with-stand external forces in various environments and thus leading the improved creep and chemical resistance.

Supplementary Figure 10. Photographs of EN-VanEHBP7 in solvents at room temperature before (a) and after (b) 72 h.

Supplementary Table 9. The swelling ratio and gel fraction of the epoxy supramolecular thermosets

Samples	Swelling ratio(%)						Gel fraction (%)
	H ₂ O	EtOH	THF (25°C)	DMF (25°C)	THF (65°C)	DMF (120°C)	
EN-DGEBA	1.6	1.5	1.4	1.9	14.7	35.3	95.0
EN-VanEP	2.5	5.0	1.5	2.1	13.6	44.0	94.1
EN-VanEHBP7	1.3	4.5	0.7	1.2	12.0	30.4	95.6

Referee comment 7. Review and clarify the results of stress relaxation experiments, particularly the apparent drop in stress below 0% for EN-VAN-HBP-7 at 130 °C after 30 s.

Response: We gratefully thank you for your instructive suggestions. Based on your suggestion, we have carefully repeated the stress relaxation experiment EN-VanEHBP7 at 130 °C (Fig. 6b). The stress of EN-VanEHBP7 decreased gradually to close to 0% at 130 °C after 30 s.

Fig. 6. (a) Rearrangement of EN-VanEHBP7 and the reprocessing process. (b) Stress relaxation curves of EN-DGEBA, EN-VanEP and EN-VanEHBP7 at different temperatures. (c) The fitted curves between $\ln t$ and $1000/T$ for epoxy supramolecular thermosets. (d) Temperature dependence of the thermal expansion of the epoxy supramolecular thermosets. (e) Tensile strength of the original and reprocessed EN-VanEHBP7. (f) Storage modulus and $\tan \delta$ curves of original and reprocessed EN-VanEHBP7. (g) Comparisons of the tensile strength and reprocessing time of thermosets containing imine bonds in this work and in previous studies.

Referee comment 8. In results, the sample name starting with “En-”, what does this mean?

Author response: Thank you for your comment. “En-” represents “epoxy supramolecular network” in this study. Accordingly, we have added more discussions about the illustration of “En-” in the revised manuscript.

Then, VanEP and VanEHBP were cured with CBA-TTE without a catalyst to form epoxy supramolecular networks, which were synergistically crosslinked by transesterification and hydrogen bonds. The resultant epoxy supramolecular network (EN-VanEHBP) exhibited excellent mechanical properties, improved creep and chemical resistance, and demonstrated fast reprocessability.

Referee comment 9. On page 4, line 83, clarify whether VanEP can be categorized as epoxy resin.

Author response: We appreciate your very good suggestion. The chemical structure of VanEP has been added in the Supplementary Information (Figure S1). According to the chemical structure of VanEP, it can be categorized as epoxy resin.

Supplementary Figure 1. Synthetic approaches for VanEP, VanEHBP and CBA-TTE.

Referee comment 10. The term 'FAT' is not explained or illustrated in either the main text or supplementary information.

Author response: We are very sorry that the inconsistent of term “FAT” and “CBA-TTE” in previous submission. The relevant revision in the revised manuscript is as follow in blue font. The **carboxyl-terminated polyetheramine (CBA-TTE)** used as a curing agent was **synthesized from 4-formylbenzoic acid and trimethylolpropane tris[poly(propylene glycol), amine terminated] ether.**

Referee comment 11. References are missing on page 3, lines 51-55, where the previous work is mentioned. Attach the relevant references for clarity.

Author response: We thank you for carefully reading and the relevant references were attached in the revised manuscript.

A supramolecular polyimine with high tensile strength and toughness was developed by introducing dynamic imine bonds and hydrogen bonds, which can depolymerized at room temperature in THF/HCl mixture solution [22]. Cross-linked polymers based on reversible amidation chemistry with maleic anhydride and secondary amine monomers can be depolymerized in HCl solution to realize recovery of monomers with high purity and yield.

[22] Zhang, Z. et al. Strong and tough supramolecular covalent adaptable networks with room- temperature closed-loop recyclability. *Adv. Mater.* **35**, 2208619 (2022).

Referee comment 12. The manuscript contains spelling mistakes, such as "recycledat" on page 3, line 48. A thorough review for such errors is necessary.

Author response: We are very sorry for the spelling errors in the text. Accordingly, we have improved the English Language by Nature Research Editing Service (The verification code 5753-151F-3CC7-DD3F-643P) in the revised manuscript.

Finally, we would like to take this opportunity to express our heartfelt gratitude once again for your valuable comments on our paper. These comments are very important for further improving the quality of our paper. Thank you very much for your positive attitude in this work, which encourages us to carry out more studies on closed-loop recycling of epoxy thermosets.

Response to Referee 2

General comments: This work constructed closed-loop recyclable tough epoxy thermosets by hyperbranched topological structure. Epoxy thermosets have been widely used in our life due to their excellent thermal and mechanical properties and solvent resistance. However, they are difficult to be recycled or reused due to their permanently cross-linked networks. The preparation of epoxy thermosets with high mechanical performance and efficient recyclability is still a challenge. In this manuscript, authors addressed these issues using one epoxy system, and vanillin-based hyperbranched epoxy resin was synthesized to prepare tough epoxy supramolecular thermosets. The obtained cross-linked epoxy thermosets exhibited outstanding mechanical properties and room-temperature degradation recyclability, and the toughening mechanism and degradation mechanism were studied and discussed in detail. This study provides an easy strategy to develop ultrastrong, tough, and recyclable epoxy thermosets. Moreover, renewable resource vanillin was used, which made this work more sustainable. This article is the result of an up-to-date research, very well designed, containing a large volume of detailed experiments, and the results are very well arranged and organized. Thus, this is an interesting work and I recommend the publication of this manuscript. Some improvement suggestions are as follows.

Author response:

We sincerely thank you for your careful reading of our manuscript and for recommending our manuscript to be published in Nature Communications after minor revision. The valuable suggestions and comments provided by the referee are greatly helpful to improve our manuscript. Below we answer the referee's questions and comments in a point-by-point basis. We hope the referee will be satisfied with the revised manuscript as well as our responses

Referee comment 1. The incorporation of VanEP, CBA-TTE and VanEHBP for epoxy thermosets could be considered as the novel part of this study. The chemical structures and abbreviations of the used materials for the synthesis of VanEP, CBA-TTE and VanEHBP should be provided.

Author response: We thank you for raising this important point. As suggested, we supplemented the chemical structures and abbreviations of VanEP, CBA-TTE and VanEHBP in the revised manuscript (Figure S1).

Supplementary Figure 1. Synthetic approaches for VanEP, VanEHBP and CBA-TTE.

Supplementary Figure 2. Chemical structure of VanEHBP.

Referee comment 2. The abbreviations of carboxyl terminated epoxy curing agent containing dynamic imine bonds should be consistent, such as CBA-TTE in Figure 3 but FAT in Figure S1.

Author response: We are sorry for the inconsistency. The relevant revisions for the inconsistent terms in the revised manuscript is as follow in blue font.

Fig. 4. (a) Synthetic routes of EN-VanEP and EN-VanEHBP. Snapshots of the all-atom MD simulations of the structures and schematic illustration of the structures of (b) EN-VanEP and (c) EN-VanEHBP, showing that hydrogen bonds are present at a greater density in EN-VanEHBP than in EN-VanEP.

Supplementary Figure 16. (a) Chemical structures and abbreviations of VanEP, CBA-TTE and VanEHBP, (b) FT-IR spectra of CBA-TTE, VanEP and VanEHBP, ¹H NMR spectra of (c) CBA-TTE, (d) VanEP, and (e) VanEHBP.

Referee comment 3. In Figure S6, the sample information should be clearly indicated.

Author response: We highly appreciate your comments. The sample information was indicated in Figure S3.

Supplementary Figure 3. Temperature-dependent FT-IR spectra of EN-VanEHBP7 during curing.

Referee comment 4. Figure 2 displayed FT-IR spectra in the regions of 3600-3000 cm^{-1} and 1800-1500 cm^{-1} , the Temperature-dependent FT-IR spectra of VanEHBP7 in the range of 4000-500 cm^{-1} should also be provided.

Author response: We fully agree with you that the Temperature-dependent FT-IR spectra of VanEHBP7 should be provided. To address this comment, we added the FT-IR spectra in the revised manuscript. The detailed text revisions are in blue font as below.

Supplementary Figure 5. Temperature-dependent FT-IR spectra of EN-VanEHBP7 upon heating from 20-150 °C.

Referee comment 5. The detail descriptions of the Positron annihilation lifetime (PAL) measurement method should be added in section of Characterization.

Author response: We sincerely thank you for your suggestion. We supplemented detailed measurement descriptions of the Positron annihilation lifetime (PAL) in the revised manuscript.

Positron annihilation lifetime (PAL) measurements were recorded by a conventional fast-fast coincidence system with a temporal resolution of approximately 210 ps. The positron source was prepared by depositing and drying a $^{22}\text{NaCl}$ aqueous solution (with an activity of ~ 500 kBq) onto the central zone (diameter of less than 2 mm) of a Kapton polyimide foil (10 mm \times 10 mm \times 7.5 μm , Nilaco) and then covering the first foil with another Kapton polyimide foil of the same size. The positron source was sandwiched between two identical samples with dimensions of 10 mm \times 10 mm \times 1 mm. The sample-source-sample set was sealed in a vacuum chamber that was evacuated by a rotary pump and a turbo molecular pump. The PAL spectrum was collected over 4096 channels with a channel width of 13.24 ps/ch. The total counts of 4×10^6 for each PAL spectrum at a fixed temperature were collected in 60 min at a counting rate of approximately 1300 counts/s. It took approximately 2460 minutes for the temperature-dependent PAL experiments for each sample, and the interval between adjacent temperature was approximately 100 minutes. To prevent the backscattering of γ -rays by the PAL spectroscopy scintillators, the two PAL spectroscopy detectors were positioned perpendicularly. The distance between the sample-source-sample set and each lifetime detector was ~ 20 mm. The vacuum inside the sample chamber was greater than 1×10^{-6} Pa. By using the PATFIT program, all PAL spectra were decomposed into three lifetime components of τ_1 , τ_2 and τ_3 ($\tau_1 < \tau_2 < \tau_3$), with corresponding intensities of I_1 , I_2 and I_3 , respectively. By using a semiempirical equation based on a spherical infinite potential well model (the Tao-Eldrup model), the average radius (R) of the free-volume holes were estimated from the o-Ps lifetime τ_3 :

$$\tau_3^{-1} = 2 \left[1 - \frac{R}{R+\Delta R} + \frac{1}{2\pi} \sin \left(2\pi \frac{R}{R+\Delta R} \right) \right] \quad (\text{S1})$$

where ΔR is the thickness of the electron layer on the surface of the free-volume holes, which is an empirical parameter of 0.1656 nm. The average volume (V_f) of free-volume holes, which is usually called the free-volume hole size, was estimated from

$$V_f = 4\pi R^3/3 \quad (\text{S2})$$

The relative fractional free-volume (in %) was defined as

$$f_r = V_f I_3 \quad (\text{S3})$$

where f_r is the free volume, I_3 is the annihilation intensity, and C is a constant (usually 0.018).

Referee comment 6. Please add the detail recovery efficiency of mechanical properties of different reprocessed and chemical recycled samples in Table S6.

Author response: We sincerely appreciate your insightful comments. The detail recovery efficiency of mechanical properties of different reprocessed and chemical recycled samples have been added to Table S5.

Supplementary Table 5. Tensile strength of original, reprocessed and chemical recycled epoxy supramolecular thermosets

Samples		EN-DGEBA	EN-VanEP	EN-VanEHBP7
Original	Tensile strength (MPa)	66.3 ± 5.6	73.0 ± 4.8	104.5 ± 3.4
	Recovery efficiency (%)	86.3	94.2	99.5
1 st Reprocessing cycled	Tensile strength (MPa)	57.2 ± 2.0	66.9 ± 1.4	104.0 ± 2.8
	Recovery efficiency (%)	86.3	94.2	99.5
2 nd Reprocessing cycled	Tensile strength (MPa)	60.7 ± 3.8	68.6 ± 1.1	104.0 ± 4.7
	Recovery efficiency (%)	91.5	94.6	99.5
3 rd Reprocessing cycled	Tensile strength (MPa)	65.5 ± 4.9	69.1 ± 1.5	104.3 ± 1.0
	Recovery efficiency (%)	98.7	95.2	99.8
Chemical recycled	Tensile strength (MPa)	54.2 ± 1.3	65.9 ± 3.1	105.5 ± 2.1
	Recovery efficiency (%)	81.7	90.3	100.5

Referee comment 7. In Figure S12, it is beneficial to give the mass variations of epoxy thermosets in different solvent.

Author response: Thank you for the comment. We have conducted supplementary experiments related to this point and provided the mass variations in the revised manuscript.

Supplementary Table 9. The swelling ratio and gel fraction of the epoxy supramolecular thermosets

Samples	Swelling ratio(%)						Gel fraction (%)
	H ₂ O	EtOH	THF (25°C)	DMF (25°C)	THF (65°C)	DMF (120°C)	
EN-DGEBA	1.6	1.5	1.4	1.9	14.7	35.3	95.0
EN-VanEP	2.5	5.0	1.5	2.1	13.6	44.0	94.1
EN-VanEHBP7	1.3	4.5	0.7	1.2	12.0	30.4	95.6

Finally, we would like to take this opportunity to express again our sincere appreciation to your valuable comments on our paper. These comments are very important for further improving the quality of our paper. Thank you very much for your positive attitude in this work which encourages us to carry out more studies on closed-loop recycling of epoxy thermosets.

Response to Referee 3

General comments: The work reports a recyclable and tough epoxy resin with mediocre performance and less remarkable recyclability in comparable works. The noteworthy aspect lies in the author's conceptualization of supramolecular thermosets, elaborating the positive effect of hyperbranched topological structure on forming H-bonds. But this could be foreseeable in this field. There is no outstanding contribution in CANs and closed-loop recycling of plastics. The innovation of the proposed strategy and material structure is insufficient, falling short to meet the high requirements of 'Nature Communications'.

Author response: We would like to sincerely thank for your thorough reading of our manuscript and your insightful input and highly valuable suggestions. Rigorous revisions were conducted according to the comments and suggestions. We believe that the supplement experimental tests and discussions have greatly improved our work. All the revised or newly added contents have been marked in blue font, in both the Response Letter and revised manuscript.

Referee comment 1. The quality of manuscript requires significant improvement. The overall expression of the whole manuscript is not clear. The structure of the article might consider extensively reorganization (native English writing, scientific and clear expression, article logicity).

Author response: Thank you for sparing time reading our manuscript and your critical comments. We have tried to provide more detailed explanation in the revised manuscript according to your comments. The English writing was improved by Nature Research Editing Service (The verification code 5753-151F-3CC7-DD3F -643P) in the revised manuscript.

Referee comment 2. The proposed strategy is acceptable, but the schematic diagram is simplistic that cannot represent the hyperbranched topological structure described in the manuscript and its potential effect on cycling of epoxy.

Author response: The referee raised a very important point concerning the effect of the hyperbranched topological structure on the cycling of epoxy. We fully agree with the referee on this point. In fact, we have considered this at the starting stage of our study. We have conducted experiments to systemically investigate the role of hyperbranched topological structure in the cycling of epoxy. We have reorganized the data and elaborated the role of hyperbranched topological structure in cycling of epoxy. In this work, the recycling of epoxy involves acid-aided hydrolysis and aldehydeamine reaction of imine bonds. The recycling process occurred in two steps: (1) degradation of thermosets in acidified solvents to generate aldehydes and protonated amines, (2) re-formation to recover thermosets while removing the solvent and acid. Therefore, the solubility of depolymerized prepolymer and healing efficiency of re-crosslinking thermosets are crucial for the recycling of epoxy. Because of the low solvation energy induced by highly branched molecular structure, the depolymerized EN-VanEHBP7 thermoset demonstrated better solubility than that of EN-DGEBA and EN-VanEP samples, resulting in highly efficient degradation of EN-VanEHBP7. In the supramolecular network of EN-VanEHBP7, the high-density multiple hydrogen bonds and

branched structure of VanEHBP prevents the ordered packing of the molecules to provide “free” moieties. During the re-crosslinking, besides aldehydeimine reaction of imine bonds, the free moieties can exchange with the associated hydrogen bonds to provide excellent healing ability and recover the strong mechanical strength of EN-VanEHBP7. Together, the hyperbranched structure of VanEHBP afford excellent recycling capability and unique strength and toughness for the epoxy supramolecular thermosets. The relevant schematic diagram was corrected in the revised manuscript.

Fig. 8. (a) Closed-loop mechanism of EN-VanEHBP7. (b) ^{13}C NMR spectra of the original, chemically recycled and liquid-state depolymerized EN-VanEHBP7. (c) Real-time ^1H NMR spectrum of the EN-VanEHBP7 degradation solution. (d) Tensile curves. (e, f) Storage modulus and $\tan \delta$ of epoxy supramolecular thermosets before and after chemical recycling. (g) Comparisons of the tensile strength and T_g of polymers that can be chemically recycled at room temperature in this work and in previous studies.

Referee comment 3. The introduction section should be extensively revised from the discussion of significant scientific issues in this field.

Author response: We sincerely thank you for your critical suggestion. We have extensively revised the introduction section to better present the scientific issues, especially the related studies about recycling of supramolecular thermosets and hyperbranched polymers for programming polymer properties, as shown below.

Epoxy thermosets that exhibit excellent mechanical and thermal performance have attracted considerable attention in the fields of coatings, adhesives, and structural components^{1,2}. However, the covalent crosslinked structure of epoxy thermosets hinder reprocessing and recycling, resulting in substantial economic and environmental problems^{3,4}. To overcome these problems, covalent adaptable networks (CANs) provide a pragmatic solution, allowing the fabrication of cross-linked healable and recyclable epoxy thermosets, which can dissociate or reversibly crosslink under certain conditions^{5,6}. To date, various dynamic covalent-bond-forming processes, including urethane exchange⁷, disulfide exchange^{8,9}, imine exchange^{10,11}, boronic ester exchange¹², and transesterification reactions^{13,14} have been employed to construct CANs, which can be used to produce reprocessable and chemically recyclable epoxy thermosets. Despite considerable effort, the reprocessing process of CANs often relies on a catalyst, a high temperature and high pressure, resulting in unwanted side reactions and a loss of physical properties^{15,16}. Consequently, the construction of CANs that are mechanically strong and thermochemically stable and can be rapidly reprocessed under mild conditions without a catalyst remains challenging. Excitingly, dynamic covalent crosslinking enables the resulting crosslinked epoxy thermosets to depolymerize into the original monomers and oligomers, which can then regenerate the thermosets^{17,18}. However, these technologies usually require high temperatures, prolonged reaction times, substantial separation and purification^{19,20}. There is a critical need to explore efficient, low-cost recycling technologies to achieve closed-loop chemical recycling through depolymerization into monomers and oligomers, followed by full repolymerization of these monomers and oligomers.

To achieve highly efficient recycling in an economically manner, more recently, several studies have reported simple and energy-saving recycling methods to develop CANs that can be recycled at room-temperature^{21,22}. For instance, a supramolecular polyimine with high tensile strength and toughness was developed by introducing dynamic imine bonds and hydrogen bonds, and this material could be depolymerized at room temperature in a THF/HCl mixed solution²². Cross-linked polymers formed via reversible amidation chemistry from maleic anhydride and secondary amine monomers can be depolymerized in HCl to realize recover their high-purity in high yield. A supramolecular polysaccharide composed of sodium alginate (SA) and cetyltrimethylammonium bromide (CTAB) was designed, and could be recycled at room temperature via water-induced plasticization²³. Nevertheless, these room-temperature chemically recyclable polymers exhibited relatively low T_g values. To improve the T_g , a recyclable poly-(diketoenamine) with high T_g (≥ 120 °C) was synthesized from triketones and aromatic or aliphatic amines and could be depolymerized at room temperature under strongly acidic conditions (5.0 M H_2SO_4)²¹. Furthermore, the existing chemical recycling processes are not yet economical because it is difficult to achieve complete repolymerization of the degradation products. To circumvent such selectivity and quantitative problems, more work is needed to improve the sustainability of the chemistry used to achieve conversion rates of 100% for the degradation product.

In most cases, epoxy thermosets based on CANs have been reported to exhibit unsatisfactory mechanical strength and thermal and oxidative stability due to their conflicting nature that limits their use in structural applications²⁴. To overcome this limitation, rigid structures with high-density crosslinking have been widely used for reversible reinforcement. High-density cross-linking can improve the mechanical strength of CANs, but is usually achieved at the expense of reprocessability and recyclability²⁵. The incorporation of reversible noncovalent interactions such as multiple hydrogen bonds into CANs can not only endow these polymers with excellent mechanical performance but also provide recyclability and self-healing capacity²⁶. In addition to hydrogen bonds, hyperbranched topological structures have also been applied to effectively improve the strength, toughness, solvent resistance and dimensional stability of CANs^{27,28}. Hyperbranched polymers have attracted considerable interest due to their unique properties and structural diversity, and have subsequently been used for various applications. Compared with conventional linear polymers, hyperbranched polymers possess intramolecular cavities and abundant functional groups, thus demonstrating excellent potential for strengthening and toughening materials^{8,18}. In particular, we fabricated a series of hyperbranched polymers with different hyperbranched topological structures including hyperbranched ionic liquids, hyperbranched epoxy resins and hyperbranched polyesters to tune the polymer properties^{17,29-31}. The topologies of these materials resulted in more efficient energy dissipation of the hyperbranched topological structures, which could simultaneously improve the strength and toughness of thermosets, as well as their functions such as flame-retardancy and recyclability^{17,29,30}. To make CANs truly suitable as replacements for traditional epoxy thermosets, a different dynamic supramolecular cross-linking approach is needed to design robust CANs that incorporate a hyperbranched topological structure to improve the mechanical durability and thermal stability of dynamic epoxy thermosets.

For these purposes, we hypothesized that if a hyperbranched topological structure that could isomerize to form CANs could be designed, this unique structure could lead to both high strength and excellent toughness. Thus, in this work, we designed a class of epoxy supramolecular thermosets, capable of rapid reprocessing and room temperature closed-loop chemical recycling. As shown in Figure 1 and Figure S1, vanillin derivatives bearing phenolic hydroxyl groups (Van-TTE) were prepared by conjugating vanillin with trimethylolpropane tris[poly(propylene glycol), amine terminated] ether. Then, Van-TTE was reacted with epichlorohydrin to obtain vanillin-based epoxy resin (VanEP). Next, vanillin-based hyperbranched epoxy resin (VanEHBP) was synthesized from VanEP and bisphenol A via proton transfer polymerization (Figure S1 and Figure S2). The carboxyl-terminated polyetheramine (CBA-TTE) used as a curing agent was synthesized from 4-formylbenzoic acid and trimethylolpropane tris[poly(propylene glycol), amine terminated] ether. Then, VanEP and VanEHBP were cured with CBA-TTE without a catalyst to form epoxy supramolecular networks, which were synergistically crosslinked by transesterification and hydrogen bonds. The resultant epoxy supramolecular network (EN-VanEHBP) exhibited excellent mechanical properties, improved creep and chemical resistance, and demonstrated fast reprocessability. More importantly, these epoxy thermosets could be converted into soluble oligomers at room temperature, and then completely regenerated into crosslinked networks without compromising their performance.

Referee comment 4. In Line 81, Regarding the paragraph ‘As shown in Figure 1, Van-TTE was formed.....’, Where is the ‘Van-TTE’ label in Figure 1? The description in text and figure should be carefully checked.

Author response: We apologize for the unclear description. We have added the ‘Van-TTE’ label in Figure 1 and carefully revised the descriptions for ‘Van-TTE’ in the revised manuscript.

Referee comment 5. From the tensile stress-strain curve, the modulus observed did not improve with the concentration of HBP (Fig.4a). Why the storage modulus below T_g yet significantly improves from DMA results (Fig.4b)? And the impact toughness of all materials should be supplied.

Author response: We gratefully thank you for your instructive suggestions. These valuable comments are greatly helpful for us to explore a tough epoxy thermosets by hyperbranched topological structure. In response to these comments, we have re-run some of the experiments and supplemented the data for tensile properties, including tensile strength and tensile modulus of all samples. As shown in Figure 5 and Table S2, the modulus increased with the addition of HBP. The improvement of modulus is due to the supramolecular networks of epoxy thermosets based on HBP. Following your suggestion, we further studied the impact toughness of all materials (Table S2). The incorporation of VanEHBP significantly improved the impact strength of epoxy thermosets. The tensile strength, toughness and impact strength of EN-VanEHBP7 were 104.5 MPa, 3.58 MJ·m⁻² and 57.0 kJ·m⁻², respectively, which were 43.1%, 109.4% and 128.0% greater than those of the cured EN-VanEP (73.0 MPa, 1.71 MJ·m⁻² and 25.0 kJ·m⁻²).

Fig. 5. (a) Stress-strain curves of epoxy supramolecular thermosets. (b) Storage modulus and (c) tan curves of epoxy supramolecular thermosets. SEM images of (d) EN-DGEBA, (e) EN-VanEP and (f) EN-VanEHBP7. (g) AFM images of EN-VanEHBP7. (h) Schematic illustration of the supramolecular networks of EN-VanEHBP upon deformation.

Supplementary Table 2. Mechanical and thermomechanical properties of the epoxy supramolecular thermosets

Samples	Tensile Strength (MPa)	Toughness (MJ·m ⁻²)	Young's Modulus (MPa)	Impact Strength (KJ·m ⁻²)	E _c (GPa)	T _g (°C)	E _d (MPa)	ρ (×10 ⁻³ mol cm ⁻³)
EN-DGEBA	66.3±5.6	1.51±0.14	1832±334	24.0±2.7	1.5	105	2.9	0.29
EN-VanEP	73.0±4.8	1.71±0.09	1945±245	25.0±3.4	1.8	83	1.3	0.14
EN-VanEHBP2	74.1±3.2	1.80±0.12	1998±281	29.5±3.6	1.7	86	1.3	0.13
EN-VanEHBP5	92.9±3.1	2.66±0.10	2332±256	48.3±3.5	2.4	92	3.0	0.30
EN-VanEHBP7	104.5±3.4	3.58±0.08	2400±214	57.0±3.3	2.8	95	4.6	0.46
EN-VanEHBP10	81.3±3.2	1.81±0.13	2223±273	42.6±3.6	2.1	86	2.8	0.29

E_c is the storage modulus at 30 °C, T_g is the glass transition temperature, E_d is the storage modulus at T_g + 30 °C, and ρ is the crosslinking density.

Referee comment 6. Close-looped cycling is doubtful. Does it specifically refer to the complete recovery of monomers. The molecular weight of the degraded solution should be supplied. In Fig.8, the description regarding the recycling of materials is confused, which should be clearly showed in the figure.

Response: We sincerely appreciate your comments regarding more in-depth discussion on the close-looped cycling. These valuable comments are of great significance to improve the quality of our manuscript. The supramolecular thermosets constructed by hyperbranched topological structure can be easily fully-depolymerized under room temperature, and then completely repolymerized without weight loss and complex separation and purification. Following your suggestion, we have performed GPC measurements with the degraded solution and reorganized the interpretation of close-looped cycling.

Fig. 8. (a) Closed-loop mechanism of EN-VanEHBP7. (b) ^{13}C NMR spectra of the original, chemically recycled and liquid-state depolymerized EN-VanEHBP7. (c) Real-time ^1H NMR spectrum of the EN-VanEHBP7 degradation solution. (d) Tensile curves. (e, f) Storage modulus and $\tan \delta$ of epoxy supramolecular thermosets before and after chemical recycling. (g) Comparisons of the tensile strength and T_g of polymers that can be chemically recycled at room temperature in this work and in previous studies.

Supplementary Figure 11. GPC analysis of the degraded solution of EN-VanEHBP7.

Referee comment 7. In the conclusion section, the claim that the EP thermoset achieved a high toughness of 3.58 MJ/m³. Checking toughness results of 3.58 MJ/m³, and the ‘m³’ should be ‘m²’.

Author response: We sincerely thank you for careful reading. The mistake in units has been corrected.

Referee comment 8. Titles in Results and Discussion seem too ordinary, just showing the properties of material or the characterizations. It is suggested to improve the overall article structure in line with the language style of ‘Nature Communications’.

Author response: We highly appreciate your comments, which are greatly helpful for us to improve the quality of our manuscript. We have revised such expressions as recommended.

Referee comment 9. The ‘Abstract’ and ‘Conclusion’ parts need to be rewritten.

Author response: Once again, we express our gratitude to you for your insightful suggestions on the manuscript. We have carefully revised the ‘Abstract’ and ‘Conclusion’ parts.

Abstract

Covalent adaptable networks (CANs) may be beneficial for the recycling and reuse of cross-linked epoxy thermosets. However, regulation of the topological structure of these networks is challenging, thus limiting the recyclability, mechanical strength and stability of epoxy CANs. Here, we report a novel and general strategy to develop strong and tough epoxy supramolecular thermosets with rapid reprocessability and room-temperature closed-loop recyclability. These supramolecular structures were constructed from vanillin-based hyperbranched epoxy resin (VanEHBP) via the introduction of intermolecular hydrogen bonds and dual dynamic covalent bonds (imine exchange and transesterification) and the formation of intramolecular and intermolecular cavities. The reversible hydrogen bonds and intramolecular and intermolecular cavities in these supramolecular thermosets endowed these materials with a remarkable energy dissipation capability, resulting in impressive ultra toughness and ultimate strength. Due to the dynamic imine exchange and reversible noncovalent crosslinks, the produced thermosets could undergo rapid and efficient

reprocessing at 120 °C within 30 s, with a nondestructive recovery rate of up to 100%. Importantly, the supramolecular thermosets could be efficiently depolymerized at room temperature, and the recovered materials maintained the structural integrity and mechanical properties of the original samples with nearly 100% material efficiency. This strategy represents a design guideline for the development of tough, closed-loop recyclable epoxy thermosets for practical use.

Conclusion

In summary, we developed a novel strategy for preparing strong and tough epoxy thermosets with supramolecular networks by introducing a vanillin-based hyperbranched epoxy resin (VanEHBP). This unique supramolecular network enabled rapid and efficient reprocessing and room-temperature closed-loop chemical recycling of epoxy supramolecular thermosets. The obtained thermosets exhibited much greater tensile strength, toughness and impact strength than traditional petroleum-based products. The supramolecular networks underwent fast stress relaxation and could be reprocessed at 120 °C within only 30 s, with nearly 100% recovery of the mechanical performance after multiple reprocessing cycles. The supramolecular networks exhibited enhanced resistance to deformation and thus improved creep and chemical resistance during service. The epoxy supramolecular thermosets could be chemically fragmented into immediately reusable monomers at room temperature, and the obtained fragment mixture could be reused for recrosslinking to reconstruct the epoxy thermosets with nearly 100% material efficiency without losing their original mechanical properties. This work provides a robust supramolecular crosslinking strategy that affords hyperbranched topological structures for designing energy-efficient and fully closed-loop recycled polymeric products.

Referee comment 10. There are so many spelling mistakes, formatting errors and unscientific statements. Some are listed as follows. The quality of manuscript should be improved at least.

a) Spelling mistakes:

Line 28, 'wok'

Line 89, 'excelent'

line 272, "depolymerised"

b) Miss spaces:

Line 32 'thermosetsthat', 'performancehave';

Line 38 'includeurethane';

Line 44 'andthere';

Line 48 'recycledat'?

Line 72, 'cavitiesand', 'instrengthening'

Line 144 'Theincrease'

Line 148 'andtended'

Line 220 'VanEHBPwhich'

c) Others:

In line 148, 'thus' should be 'Thus';

line 122, "simulated" should be "stimulation"

Supporting information:

Such as S1.3: '181 mgKOH•g⁻¹'; '1H NMR (400MHZ, DMSO-D6)'

Author response: We are very sorry for the spelling errors in the text. We have corrected the typos and poorly worded sentence in the revised manuscript and we have improved the English Language by Nature Research Editing Service (The verification code 5753-151F-3CC7-DD3F-643P) in the revised manuscript.

We would like to take this opportunity to express again our sincere appreciation to your valuable comments on our paper. These comments are very important for further improving the quality of our paper. Thank you very much for your positive attitude in this work which encourages us to carry out more studies on closed-loop recycling of epoxy thermosets.

REVIEWER COMMENTS

Reviewer #1 (Remarks to the Author):

The authors have made some improvements to this manuscript. However, the following issues still need to be addressed and explained:

1. How to calculate toughness in Supplementary Table 2? The method should be mentioned in SI. From current data, the mechanical properties of VanEHBP cannot be addressed as “impressive ultra toughness”.
2. What is the reason for the difference in mechanical properties of EN-VanEHBP7 and EN-VanEHBP10? From Supplementary Table 2, it seems like one with higher cross-linked density, as a result, one has higher tensile stress and toughness. What causes the EN-VanEHBP7 to have the highest crosslinking density?
3. The authors have reconducted the stress relaxation experiment of EN-VanEHBP7. However, the recalculated apparent activation energy is 26.1 kJ/mol, which is inconsistent with the previous data (17.3 kJ/mol). What is the error bar of this data? The stress relaxation experiment of EN-DGEBA and EN-VanEP should be repeated and the error bar should be mentioned.
4. How to measure and pick up the T_g point in your experiment? From Fig 1d, the glass transition point is not clear, there is no significant difference between EN-VanEHBP7 and EN-VanEHBP10. Why is the ordinate units of Fig 1d is strain (%) ?
5. Fig 1g is meaningless and misleading. For different materials, the recycling temperature is completely different. Using this figure to present the fast reprocessability of materials is improper.
6. Please explain the difference between epoxy and epoxy resin, and check the expression “VanEP” in the manuscript.

Therefore, I still cannot recommend this manuscript for publication in Nature Communications in the present form.

Reviewer #2 (Remarks to the Author):

The authors addressed all the issues properly. The manuscript can be accepted now.

Reviewer #3 (Remarks to the Author):

After carefully reviewing the revised manuscript and considering the improvements made by the authors in response to the reviewer's feedback, the authors have addressed several key points raised during the review and made necessary improvement regarding the whole text. The conceptualization of supramolecular thermosets remains a noteworthy aspect of the study. I am also interested in the hyperbranched topological structures on cycling epoxy resin. The innovative approach presented by the authors contributes to advancing understanding in the field. Minor revisions are suggested for further improvement, but overall, the manuscript aligns with the standards expected for publication in 'Nature Communications'.

1. Some descriptions of closed-loop recycling can be more scientific.
2. Lack of some experimental materials should be supplied, also, the purity of chemicals.
3. I wonder if there are still permanently cross-linked sites (undegradable sites?) in your closed-loop cycled epoxy network structure?

Response to Referee 1

General comments:

The authors have made some improvements to this manuscript. However, the following issues still need to be addressed and explained.

Author response: Thank you very much for your positive comments on our revision. We have carefully considered the issues that you mentioned and revised accordingly.

Referee comment 1. How to calculate toughness in Supplementary Table 2? The method should be mentioned in SI. From current data, the mechanical properties of VanEHBP cannot be addressed as “impressive ultra toughness”.

Author response: Thank you very much for pointing out this problem. We have added the method of toughness calculation in S1 and replaced “impressive ultra toughness” with “high toughness”. We hope this description is appropriate.

Toughness (τ) of the samples can be calculated from the area under the stress (σ)-strain (ε) curves, using the following equation:

$$\tau = \int_{\varepsilon=0}^{\varepsilon=\varepsilon_{\max}} \sigma d\varepsilon \quad (S1)$$

where ε_{\max} is the elongation at break.

Referee comment 2. What is the reason for the difference in mechanical properties of EN-VanEHBP7 and EN-VanEHBP10? From Supplementary Table 2, it seems like one with higher cross-linked density, as a result, one has higher tensile stress and toughness. What causes the EN-VanEHBP7 to have the highest crosslinking density?

Author response: Thank you very much for your careful reading of our manuscript and the inspiring question. According to our experimental results, the simultaneous improvement in both strength and toughness of epoxy supramolecular thermosets can be attributed to the synergistic effect of hydrogen bonding interactions, free volume properties variation, rigidity of chemical structure, good compatibility and high crosslinking density. The mechanical properties of EN-VanEHBP first improve and then decrease with an increasing VanEHBP content, reaching the maximum value at EN-VanEHBP7 with an intermediate content.

With the addition of VanEHBP which has many intramolecular cavities, an increase in the f_r value of the epoxy network has been achieved. In addition, epoxy chain segments crosslinked with VanEHBP extended in all directions, forming intermolecular cavities, which also contributed to the increase in f_r . However, the intermolecular hydrogen bonds restrict the chain mobility, resulting in a decrease in f_r . The PAL results (Table S4, Figure S7) indicate that the f_r value decreased first and then increased with increase in VanEHBP content, and EN-VanEHBP7 exhibited the lowest f_r value. According to our previous works [1, 2], the mechanical performance of epoxy thermosets toughening with hyperbranched epoxy resin reached their maxima at minimum free volume of the samples. Therefore, the mechanical performance of EN-VanEHBP first increases and then decreases with increasing VanEHBP content, causing the highest tensile stress and toughness to be achieved by EN-VanEHBP7.

Supplementary Figure 7. Variations in (a) tensile strength and (b) toughness as functions of relative fractional free-volume f_r .

The crosslinking density of EN-VanEHBP first increases and then decreases with increasing VanEHBP content, and EN-VanEHBP7 has the highest crosslinking density. The strong intermolecular interactions caused by VanEHBP increase the crosslinking density of EN-VanEHBP. The epoxy group of VanEHBP has a higher degree of functionality than that of linear VanEP, which also contributes to the increased crosslinking density of EN-VanEHBP. But the non-crosslinkable cavities and flexible chain segments results in a decrease in crosslinking density of EN-VanEHBP. Thus, EN-VanEHBP7 achieves the highest crosslinking density, further substantiating its excellent mechanical performance.

[1] Liu, X. et al. Ultrastrong and high-tough thermoset epoxy resins from hyperbranched topological structure and subnanoscaled free volume. *Adv. Mater.* **36**, 2308434 (2024).

[2] Wei, F. et al. Closed-loop recycling of tough and flame-retardant epoxy resins. *Macromolecules.* **56**, 5290-5305 (2023).

Referee comment 3. The authors have reconducted the stress relaxation experiment of EN-VanEHBP7. However, the recalculated apparent activation energy is 26.1 kJ/mol, which is inconsistent with the previous data (17.3 kJ/mol). What is the error bar of this data? The stress relaxation experiment of EN-DGEBA and EN-VanEP should be repeated and the error bar should be mentioned.

Author response: We appreciate your valuable suggestion on the experimental uncertainties. In the previous calculated activation energy data, each reported value was from only one specimen, and there was an apparent drop in stress below 0 % for EN-VanEHBP7 at 130 °C after 30 s. Indeed, the stress could not drop below 0 %, but it could decrease gradually to approach 0 %. According to your instructive suggestions, we have carefully repeated all the stress relaxation experiments and a minimum of four specimens were tested for each composition. Thanks to your suggestion, the data availability of activation energy has improved. Error bars were added to represent the standard deviations of the mean values obtained from the stress relaxation experiments. The stress relaxation experiment data of EN-DGEBA, EN-VanEP and EN-VanEHBP7 are presented in Table S5. We believe that with this revision it is easier to compare the activation energy of EN-DGEBA, EN-VanEP and EN-VanEHBP7.

Supplementary Table S5. The stress relaxation experimental data of the epoxy supramolecular thermosets

Samples	Relaxation time (s)				E_a (kJ·mol ⁻¹)	Average value E_a (kJ·mol ⁻¹)
	130 °C	120 °C	110 °C	100 °C		
EN-DGEBA	18.0	22.0	31.5	42.2	44.8	47.5 ± 2.8
	16.1	21.3	30	43.6	50.3	
	17.8	22.1	33.5	44.1	47.2	
	17.4	21.7	32.0	43.3	47.7	
VanEP	14.1	21.4	26.3	40.6	40.6	45.5 ± 4.9
	15.2	20.3	27.5	40.1	45.1	
	14.5	20.5	26.1	39.0	48.3	
	15.0	20.1	27.2	38.0	48.1	
EN-VanEHBP7	12.0	14.5	17.0	20.3	26.1	27.4 ± 2.7
	12.4	15.1	18.1	22.1	28.6	
	11.6	15.0	17.9	21.5	30.1	
	12.3	14.9	18.0	22.1	24.9	

Referee comment 4. How to measure and pick up the T_g point in your experiment? From Fig 1d (Fig. 6d), the glass transition point is not clear, there is no significant difference between EN-VanEHBP7 and EN-VanEHBP10. Why is the ordinate units of Fig 1d (Fig. 6d) is strain (%) ?

Author response: Thank you very much for your comments on the T_g data in Fig. 6d, specifically, the glass transition point is not clear from Fig. 6d. The T_g results in Fig. 5c and Fig. 6d were obtained in two independent experiments. In this work, we have picked up the T_g values from $\tan \delta$ curves in Fig. 5c and the T_v values from the dilatometry experiment in Fig. 6d. Fig. 5c displays $\tan \delta$ curves of epoxy supramolecular thermosets including EN-VanEHBP7 and EN-VanEHBP10. The peak of the $\tan(\delta)$ plot, signifying the T_g of the material, can be easily and clearly detected for each sample [1], as well as the changes in T_g .

Fig. 5 (c) Tan curves of epoxy supramolecular thermosets.

[1] Ran, Y. et al. Tunable backbone-degradable robust tissue adhesives via in situ radical ring-opening polymerization. *Nat. Commun.* **14**, 6063 (2023).

Fig. 6d shows dilatometry experiments for temperature dependence of the thermal expansion of the epoxy supramolecular thermosets. The volumetric strain of CANs shows a discontinued temperature dependence (similar to thermal expansion) during topology freezing transition, indicating a pseudo-secondary phase transition similar to the glass transition [1]. This so-called volumetric response is measured as the temperature-dependent axial strain using DMA. The data sections before and after the transition are fitted by linear functions. The relative Volume-Temperature characteristics, thus, depicts two transitions: the classical T_g and T_v [2]. Conventionally, T_v is treated as the onset of deviation between the low temperature linear regime and the actual curve, and this onset method is influenced by the selection of the linear range.

Figure Dilatometry experiment and determination of T_v (*ACS Macro Lett.* **1**, 789-792 2012).

[1] Capelot, M. et al. Catalytic control of the vitrimer glass transition. *ACS Macro Lett.* **1**, 789-792 (2022).

[2] Luo, J. et al. Elastic vitrimers: beyond thermoplastic and thermoset elastomers. *Matter.* **5**, 1391-1422 (2022).

Referee comment 5. Fig 1g (Fig. 6g) is meaningless and misleading. For different materials, the recycling temperature is completely different. Using this figure to present the fast reprocessability of materials is improper.

Author response: Thank you very much for pointing out this problem. According to your kind suggestion, we have removed Fig. 6g from the revised manuscript.

Referee comment 6. Please explain the difference between epoxy and epoxy resin, and check the expression “VanEP” in the manuscript.

Author response: We are grateful for your concern on the difference between epoxy and epoxy resin. The term epoxy is a prefix referring to a bridge consisting of an oxygen atom bonded to two other atoms already united in some way [1]. Epoxy resin is defined as any molecule containing one or more 1, 2-epoxy groups [1]. The term epoxy resin is applied to both prepolymers and to cured resin; the former contains reactive epoxy groups, hence their name. In the cured resin all of the reactive groups may have reacted, so that although they no longer contain epoxy groups the cured resin is still called epoxy resin [2]. We hope this response address your concerns and we have carefully checked the expression “VanEP” in the manuscript.

[1] May, C. A. *Epoxy resins: chemistry and technology*, 2nd edn (Dekker, New York, 1976).

[2] Ellis, B. *Chemistry and technology of epoxy resins*, 1st edn (Chapman & Hall, Netherlands,

1993).

We are grateful to you for your careful review and constructive feedback, which have greatly contributed to the improvement of the manuscript. We also appreciate your comment regarding the valence state change and your kind understanding on our explanation. Once again, we extend our deepest thanks to you for your constructive suggestion and guidance throughout this review process. Your comments and suggestions are valuable for us to further improve this work.

Response to Referee 2

General comments: The authors addressed all the issues properly. The manuscript can be accepted now.

Author response: We sincerely thank you for recommending our manuscript to be published in Nature Communications. We are grateful to you for giving all the valuable suggestions and comments for this work.

Response to Referee 3

General comments: After carefully reviewing the revised manuscript and considering the improvements made by the authors in response to the reviewer's feedback, the authors have addressed several key points raised during the review and made necessary improvement regarding the whole text. The conceptualization of supramolecular thermosets remains a noteworthy aspect of the study. I am also interested in the hyperbranched topological structures on cycling epoxy resin. The innovative approach presented by the authors contributes to advancing understanding in the field. Minor revisions are suggested for further improvement, but overall, the manuscript aligns with the standards expected for publication in 'Nature Communications'.

Author response: We greatly appreciate your encouraging feedback and insightful comments, and we are extremely thankful for the overall positive evaluation of our work. Your constructive comments and valuable suggestion are greatly helpful to improve the quality of our work. Thank you very much for your positive evaluation and the recommendation for publication.

Referee comment 1. Some descriptions of closed-loop recycling can be more scientific.

Author response: We sincerely appreciate the feedback on improving the clarity and readability of our work. We conducted a thorough revision of the entire text to address the descriptions of closed-loop recycling for maintaining high standards of accuracy and clarity. The changes are marked in blue in the revised manuscript.

Referee comment 2. Lack of some experimental materials should be supplied, also, the purity of chemicals.

Author response: We appreciate for this valuable suggestion. Following your suggestion, we have added the purity of chemicals in the text. The manuscript has been revised as follows.

Vanillin (Van, 99.0%), 4-formylbenzoic acid (p-CBA, 98%), trimethylolpropane tris[poly(propylene glycol), amine terminated] ether (TTE, $M_n = 440 \text{ g}\cdot\text{mol}^{-1}$), tetrabutylammonium bromide (99%), sodium hydroxide, bisphenol A (BPA, 99%), were purchased from Shanghai Macklin Biochemical Co., Ltd. Methanol ($\geq 99.7\%$), ethanol ($\geq 99.7\%$), ethylacetate ($\geq 99.5\%$), epichlorohydrin (ECH, $\geq 99.5\%$), tetrahydrofuran (THF, $\geq 99.5\%$), N,N-dimethylformamide (DMF, $\geq 99.5\%$) and hydrochloric acid (HCl, 36.0-38.0%) were purchased from Sinopharm Chemical Reagent Co., Ltd. Diglycidyl ether of bisphenol-A (DGEBA) epoxy resin (a bisphenol epoxy with epoxide equivalent weights 196 g/eq, $\geq 99.5\%$) was obtained from Baling Petrochemical Company, Inc. China. All chemicals were used as received unless described otherwise below.

Referee comment 3. I wonder if there are still permanently cross-linked sites (undegradable sites?) in your closed-loop cycled epoxy network structure?

Author response: We thank the reviewer for the critical comment. In this work, the cross-linked sites in epoxy supramolecular thermosets were formed from carboxyl groups of CBA-TTE reacted with the epoxy groups of VanEP and VanEHBP. In acidified solvents at

room temperature, imine bonds in epoxy supramolecular thermosets were dissociated by acid-aided hydrolysis, generating oligomers containing aldehyde and amine groups. In this room temperature recycling process, we expect that there are still permanently cross-linked sites as undegradable in the depolymerized oligomers. After removing the acid solutions, the re-fabricated epoxy supramolecular thermosets were realized through aldehyde-amine reaction. Thus, although there may still be permanently cross-linked sites (undegradable sites) in our closed-loop cycled epoxy network structure, the supramolecular thermosets could be efficiently depolymerized at room temperature, and then re-crosslinked to maintain the structural integrity and mechanical properties of the original samples.

We are grateful to you for your careful review and constructive feedback, which have greatly contributed to the improvement of our manuscript. Once again, we extend our deepest thanks to you for your constructive suggestion and guidance throughout this review process.

REVIEWERS' COMMENTS

Reviewer #1 (Remarks to the Author):

The revised manuscript can be accepted.